# Accessing Vision Foundation Models via ImageNet-1K

**Yitian Zhang**[1]  **Xu Ma**[1]  **Yue Bai**[1]  **Huan Wang**[1]* **Yun Fu**[1,2]
[1]Department of Electrical and Computer Engineering, Northeastern University
[2]Khoury College of Computer Science, Northeastern University
`{zhang.yitian, ma.xu1, bai.yue, wang.huan}@northeastern.edu`
`yunfu@ece.neu.edu`

## Abstract

Vision foundation models are renowned for the generalization ability due to massive training data. Nevertheless, they demand tremendous training resources, and the training data is often inaccessible, e.g., CLIP, DINOv2, posing great challenges to developing derivatives that could facilitate the research. In this work, we offer a very simple and general solution, named *Proteus*, to distill foundation models into smaller equivalents on ImageNet-1K without access to the original training data. Specifically, we remove the designs from conventional knowledge distillation settings that result in dataset bias and present three levels of training objectives, i.e., token, patch, and feature, to maximize the efficacy of knowledge transfer. In this manner, Proteus is trained at ImageNet-level costs with surprising ability, facilitating the accessibility of training foundation models for the broader research community. When leveraging DINOv2-g/14 as the teacher, Proteus-L/14 matches the performance of the Oracle method DINOv2-L/14 (142M training data) across 19 benchmarks and outperforms other vision foundation models including CLIP-L/14 (400M), OpenCLIP-L/14 (400M/2B) and SynCLR-L/14 (600M) with a significantly smaller training set of 1.2M images. Code is available at here.

## 1 Introduction

By leveraging extensive pre-training on diverse and massive datasets, vision foundation models (Oquab et al., 2023; Tian et al., 2023; Radford et al., 2021; Cherti et al., 2023) represent a significant advancement in the field of computer vision, aiming to learn comprehensive and versatile visual features that can generalize well across various downstream tasks, such as classification, segmentation, etc. As a result, vision foundation models are becoming fundamental components in the toolkit of modern computer vision research and development.

While those models have released their weights for public usage, there are limited model choices that might not cater to all scenarios, such as the application on edge devices. However, training foundation models remains largely inaccessible for most researchers due to two primary factors: (1) the training data for these foundation models is seldom disclosed. While there have been attempts to reproduce CLIP (Radford et al., 2021) using alternative datasets (Cherti et al., 2023), replicating the training of foundation models like DINOv2 (Oquab et al., 2023) and SynCLR (Tian et al., 2023) remains less explored due to private data sources. (2) even if the training data were accessible, training on these enormous datasets necessitates substantial computational resources, which may be beyond the reach of most researchers. ImageNet-1K (Deng et al., 2009), which has long been the cornerstone for advancements in the supervised learning domain, is now less frequently utilized as a training set in the era of foundation models due to its relatively 'small' scale. In this work, we seek to address the following question: ***Can the success of vision foundation models be replicated on much smaller datasets, such as ImageNet-1K, without sacrificing their generalization ability?*** In other words, we work on the problem of task-agnostic model compression with limited data.

Intuitively, leveraging the pre-trained weights of these foundation models is essential to accomplish this task and there are two possible directions: (1) recent advances (Ma et al., 2023; Xia et al., 2023)

---

*Corresponding Author.

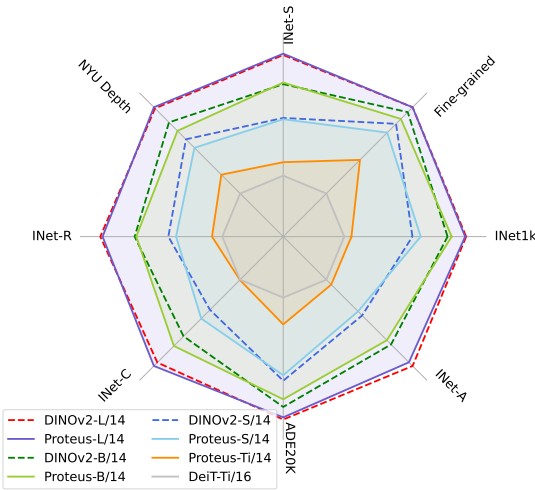

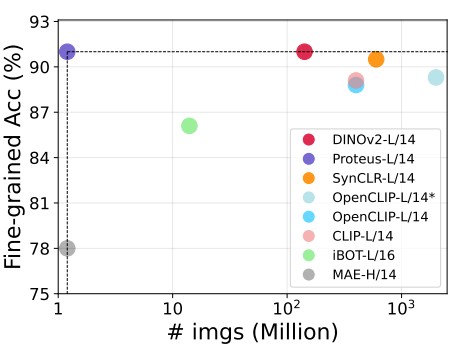

Figure 1: Distilling from DINOv2 with only 1.2M images, Proteus matches the Oracle models' performance across 19 benchmarks at different scales and our delivered ViT-Tiny model remarkably outperforms the traditional distillation method DeiT on all metrics.

Figure 2: Relationship between the number of training images and average accuracy in 12 fine-grained classification datasets. Distilling from DINOv2-g/14, Proteus-L/14 outperforms OpenCLIP-L/14* (2B) using 0.06% of its training data and matches the performance of the Oracle model DINOv2-L/14. The X-axis is on a logarithmic scale.

in Natural Language Processing (NLP) have validated that structure pruning could be a possible answer as it inherits most of the knowledge from the foundation model. However, they either sample subsets from the original enormous training set (Xia et al., 2023) or utilize high-quality, representative datasets (Ma et al., 2023) such as Alpaca (Taori et al., 2023), making it difficult to trace the success back due to the limited dataset availability in computer vision. More importantly, structure pruning requires delicate hand-crafted designs and cannot be easily generalized to arbitrary architectures, making it challenging to meet the diverse needs of real-world scenarios. (2) TinyCLIP (Wu et al., 2023) leverages knowledge distillation to transfer the fruitful knowledge from the foundation model CLIP (Radford et al., 2021) to a customized structure. Although showing great generalization ability across various datasets, it still requires the original large-scale dataset LAION-400M (Schuhmann et al., 2021) to perform task-agnostic distillation, necessitating extensive training resources and consumption. In pursuit of a more general design, we adventurously choose knowledge distillation as the means to achieve this objective, i.e., transferring the fruitful knowledge embedded in the foundation model to a randomly initialized student network.

There remain two critical issues for the knowledge transfer on ImageNet-1K: (1) the exact distributions of those undisclosed datasets, e.g., WIT400M (Radford et al., 2021), and LVD-142M (Oquab et al., 2023), are unknown and it is likely that a distribution shift exists between ImageNet-1K and those gigantic datasets. This poses significant challenges to the generalization ability of the target model, as the network tends to memorize the training images in a fixed pattern, leading to dataset bias (Torralba & Efros, 2011; Tommasi et al., 2017; Liu & He, 2024). (2) most vision foundation models (Oquab et al., 2023; Radford et al., 2021; Tian et al., 2023) are trained with self-supervised learning objectives, which require large amounts of data to be effective. Consequently, directly adopting their optimization strategies may not yield optimal results in our context.

To address the aforementioned challenges, we present *Proteus*[1], a simple and general distillation framework to transfer the fruitful knowledge from vision foundation models by emulating their behaviors. We first combat dataset bias and find the introduction of one-hot labels and the projection head in the conventional knowledge distillation setting (Hinton et al., 2015; Touvron et al., 2021) will lead to dataset bias. Consequently, we perform distillation on the intermediate features (Adriana et al., 2015) and discard the labels. To maximize the knowledge transfer power between the foundation model and the target network, we construct the proxy task by combining the token-level, patch-level, and feature-level learning objectives to learn the general-purpose visual representations, ensuring the performance of Proteus across various tasks.

---

[1]Proteus, a sea god in Greek mythology, is renowned for his shape-shifting ability to mimic various creatures. We named our method after him due to its learning pattern and generalization ability.

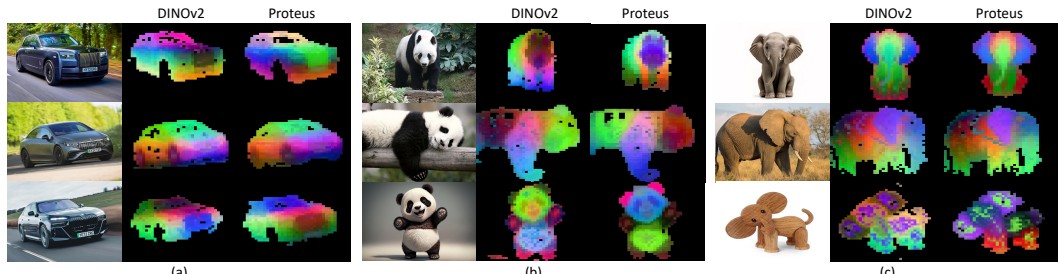

Figure 3: PCA visualization. DINOv2-B/14 is distilled from DINOv2-g on LVD-142M and Proteus-B/14 is distilled from DINOv2-L on ImageNet-1K. Details can be found in Sec. A.10.

Shown in Fig. 1 and Fig. 2, when leveraging DINOv2 (Oquab et al., 2023) as the teacher model, Proteus achieves comparable performance to the Oracle method across 19 benchmarks, which cover the tasks of classification, semantic segmentation, and depth estimation, and it outperforms other vision foundation models with significantly less data. After demonstrating the ability to match the performance of DINOv2, we further scale down the model size and introduce Proteus-Tiny, a model with a size that is unsupported by most foundation models. Fig. 1 shows that it exhibits significant improvement over the supervised distillation method DeiT (Touvron et al., 2021) on all metrics using the same training set, promising its potential as a new training paradigm for existing approaches. Owing to our general design, Proteus can be easily adapted to existing vision foundation models to access them at significantly reduced computational costs. Its generalization capability is further validated by successfully compressing SynCLR, trained on an undisclosed set of 600 million synthetic images, and CLIP, trained on the WIT-400M dataset. Abundant analysis demonstrates that Proteus is robust across various subsets of ImageNet-1K and even achieves respectable performance when limited to access to a single image.

## 2 METHODOLOGY

In this section, we present Proteus, a simple and general framework to access vision foundation models with 'limited' data, i.e., ImageNet-1K. We first introduce our efforts in mitigating the dataset bias of the proxy dataset so that Proteus can effectively transfer the general representation from the pre-trained foundation models by mimicking its behaviors. Then, we present our proxy task which encompasses multiple levels of learning objectives to promise the applications on various tasks.

### 2.1 PROXY DATASET

While we do not introduce any new dataset to try to reproduce the results of foundation models (Cherti et al., 2023), we focus on leveraging publicly available resources, e.g., ImageNet-1K, to access the vision foundation models. This task is challenging because the distribution of those large-scale datasets is usually unknown and it is very likely that there exists distribution shifts between ImageNet-1K and those private datasets. Hence, it is critical to reduce the dataset bias (Torralba & Efros, 2011) of ImageNet-1K so that our learned representation can be general enough.

**Combating Dataset Bias.** Given a pre-trained teacher network $F'(\cdot)$, a foundation model in our context, conventional knowledge distillation (Hinton et al., 2015; Touvron et al., 2021) optimizes the student network $F(\cdot)$ by two loss terms. The first one is Cross-Entropy loss which is calculated on the logits of student network $p$:

$$\mathcal{L}_{CE} = -\sum_{k=1}^{K} \hat{y}_k \log(p_k),$$ (1)

where $\hat{y}_k$ is the one-hot label for class $k$ and $K$ denotes the number of total classes. While the second one is KL divergence loss (Kullback, 1997) which enforces the prediction of student network $p$ and teacher model $p'$ to be as similar as possible:

$$\mathcal{L}_{KL} = -\sum_{k=1}^{K} p_k' \log\left(\frac{p_k}{p_k'}\right).$$ (2)

Combining the two losses uniformly, the student network $F(\cdot)$ can be updated by:

$$\mathcal{L} = (1 - \lambda) \cdot \mathcal{L}_{CE} + \lambda \cdot \mathcal{L}_{KL}, \tag{3}$$

where $\lambda$ is a hyperparameter introduced to balance the two terms. Empirically, the default setting works well in the supervised learning setting as it delivers good performance on ImageNet-1K.

However, we argue that this setup will hinder the knowledge transfer for two reasons: (1) The Cross-Entropy (CE) loss, which leverages the information of one-hot labels, can lead to dataset bias as the model tends to memorize the training images and classes. This memorization makes it challenging for the model to generalize to unseen classes during downstream evaluation. (2) The generation of class logits $p$ implicitly introduces dataset bias because the intermediate features are projected onto a pre-defined dimensionality, such as 1000 for ImageNet-1K, which may be discarded during downstream evaluation. Based on these concerns, we perform knowledge distillation before the project head (FC layer) and leverage the last-layer feature $v$ for knowledge transfer (Adriana et al., 2015).

## 2.2 PROXY TASK

Foundation models such as DINOv2 (Oquab et al., 2023) are designed to learn general-purpose visual features, excelling not only in high-level classification tasks but also in dense prediction tasks like semantic segmentation. To maximize the knowledge transfer power and promise the application on various tasks, we conduct distillation across three different levels of training objectives, i.e., token-level, patch-level, and feature-level, to transfer the fruitful knowledge by emulating the teacher's behaviors.

**Token-level Objective.** To learn the discriminative features for high-level understanding, we minimize the L2 distance to align the classification token between the teacher and student model. Specifically, the student classification token $v^{\texttt{cls}}$ will first be mapped to a higher dimension to match the channel number of the teacher's classification token $v'^{\texttt{cls}}$:

$$\hat{v}^{\texttt{cls}} = P^{\texttt{cls}}\left(v^{\texttt{cls}}\right), \tag{4}$$

where $P^{cls}(\cdot)$ denotes the projection head for the classification token and we simply adopt the combination of Layer Normalization (Ba et al., 2016) and a linear layer. Then, we enforce the model to mimic the teacher's prediction $v'^{\texttt{cls}}$ via Mean Squared Error (MSE) loss:

$$\mathcal{L}_{token} = \|\hat{v}^{\texttt{cls}} - v'^{\texttt{cls}}\|_2^2. \tag{5}$$

**Feature-level Objective.** Although the token-level learning objective alone serves as a good proxy task to obtain the discriminative visual features, it cannot guarantee decent performance on dense prediction tasks like semantic segmentation or depth estimation (see analysis of Tab. 2). To mitigate this issue, we perform feature-level knowledge transfer in a similar manner:

$$\mathcal{L}_{feat} = \|\hat{v} - v'\|_2^2, \tag{6}$$

where $\hat{v}$ is obtained by the projection head $P^{feat}(\cdot)$ for feature distillation.

**Patch-level Objective.** To further uncover the hidden knowledge from the foundation model, we construct a patch-level learning objective inspired by the idea of masked image modeling (Bao et al., 2021; He et al., 2022; Wei et al., 2022; Zhou et al., 2021; Baevski et al., 2022). Given an image $x$, an additional view $x_{mask}$ will be generated where the patches are randomly masked and it will be sent to the student network to create the intermediate feature $v_{mask}$. Following the previous procedures, we enforce the student to recover the masked regions by:

$$\mathcal{L}_{patch} = \|\hat{v}_{mask}^{patch} - v'^{patch}\|_2^2, \tag{7}$$

where the patch tokens of the foundation model $v'^{patch}$ is produced by the unmasked view $x$ and $\hat{v}_{mask}^{patch}$ is derived from the projection head $P^{patch}(\cdot)$.

Combining the three learning objectives, we construct the proxy task of Proteus as follows:

$$\mathcal{L} = \lambda_{token} \cdot \mathcal{L}_{token} + \lambda_{feat} \cdot \mathcal{L}_{feat} + \lambda_{patch} \cdot \mathcal{L}_{patch}, \tag{8}$$

where $\lambda_{token}, \lambda_{feat}, \lambda_{patch}$ are introduced hyperparameters to balance the three terms and we simply set them to 1 in our implementations without additional fine-tuning.

## 3 EMPIRICAL VALIDATION

In this part, we conduct pre-training on ImageNet-1K (Deng et al., 2009) and validate our methods under different setups. Distilling from DINOv2 (Oquab et al., 2023), we first conduct ablation to verify our designs. Then, we evaluate our method on the object recognition task and perform linear probing on ImageNet-1K and 12 fine-grained datasets. Further, we validate our approach on dense prediction tasks, including Semantic Segmentation and Depth Estimation. Moreover, we examine the scaling behavior of our method by switching backbones with different model capacities. In the end, we test the generalization ability of our method by distilling SynCLR (Tian et al., 2023) and CLIP (Radford et al., 2021), and conduct analyses on the proxy dataset.

### 3.1 EXPERIMENTAL SETUP

We conduct pre-training on the training set of ImageNet-1K (Deng et al., 2009), comprising approximately 1.2 million images distributed across 1,000 categories. We follow the training recipe of DeiT (Touvron et al., 2021) for 300-epoch training and conduct the experiments on 8 A100 GPUs with a total batch size of 1024, except the ViT-L experiment with a batch size of 256 due to GPU memory constraints. In default, Proteus is distilled from the foundation model at a larger scale with the same patch size. Following the setups in DINOv2 (Oquab et al., 2023) and SynCLR (Tian et al., 2023), we evaluate our approach on classification tasks (ImageNet-1K and 12 fine-grained classification datasets) and dense prediction tasks (semantic segmentation and depth estimation).

### 3.2 MAIN PROPERTIES

In this section, we conduct analyses on the proxy task to verify the effectiveness of our design. We first examine the effectiveness of removing the designs that lead to dataset bias. Then, we conduct ablation to validate the proposed learning objectives and compare Proteus with DeiT at all scales.

Table 1: Ablation study on combating dataset bias. ViT-S/14 models are distilled from DINOv2-B on ImageNet-1K for 300 epochs.

| Source | KL | CE | MSE | ImageNet | Fine-grained |
|---|---|---|---|---|---|
| Hard Logits | ✔ | ✔ | | 82.8 | 78.7 |
| Soft Logits | ✔ | ✔ | | 80.3 | 71.5 |
| Hard Logits | ✔ | | | 81.7 | 79.7 |
| Soft Logits | ✔ | | | 82.3 | 80.5 |
| Hint | | | ✔ | 81.7 | 85.3 |

Table 2: Ablation study on learning objectives. ViT-S/14 models are distilled from DINOv2-B on ImageNet-1K for 300 epochs.

| Token | Feature | Patch | ImageNet | Fine-grained | ADE20K |
|---|---|---|---|---|---|
| ✔ | | | 81.7 | 85.3 | 44.0 |
| | ✔ | | 79.2 | 83.3 | 47.0 |
| ✔ | ✔ | | 81.7 | 86.1 | 47.4 |
| ✔ | ✔ | ✔ | 81.8 | 85.8 | 50.0 |

**Combating Dataset Bias.** To study the importance of mitigating dataset bias, we start with the conventional knowledge distillation setting of ViT (Touvron et al., 2021), which leverages hard logits distillation and combines the Cross-Entropy loss with the KL-divergence loss (Kullback, 1997). Shown in Tab. 1, the default distillation setting of DeiT delivers very good ImageNet accuracy. However, its generalization ability (fine-grained classification) remains at the same level as distillation from ImageNet pre-trained teacher (78.7% versus 77.8%), indicating that it does not fully utilize the fruitful knowledge of the foundation teacher model. We then empirically change hard logits to soft logits and remove the CE loss as the one-hot label will introduce the dataset bias. The results support our hypothesis as soft logits distillation without CE loss delivers the highest accuracy on fine-grained classification among the four choices. Further, we revise the design to distilling before the FC layer as it implicitly introduces the dataset bias, which brings significant improvement to generalizability.

**Learning Objective.** Although hint distillation already achieves very good results on classification tasks, it is not enough to fully transfer the knowledge of foundation models as the performance gap on semantic segmentation is quite obvious (44.0% versus 51.0%). The explanation is that classification tasks only utilize the classification token which has been distilled from the teacher, whereas semantic segmentation requires the whole feature for dense prediction. Shown in Tab. 2, the performance is boosted both on classification and dense prediction after adding the feature-level learning objective. Considering that DINOv2 (Oquab et al., 2023) is trained with patch-level learning objective (Zhou et al., 2021), we mimic this behavior to enforce the student to predict the tokenized representation of the masked patch and it delivers very good performance across various tasks.

Table 3: Comparison with supervised learning from various dimensions. We compare the supervised training scheme of ViT with Proteus in terms of ImageNet accuracy, robustness, generalization ability, and dense prediction performance. All DeiT results are obtained by knowledge distillation, except * because the introduced distillation token can not fit into the backbone for downstream evaluation.

| Method | Arch | Teacher | Accuracy | Robustness | | | | Generalizability | Dense Prediction | |
|---|---|---|---|---|---|---|---|---|---|---|
| | | | Im-Val | Im-S | Im-A | Im-R | Im-C↓ | Fine-grained | *ADE20K | *NYUd↓ |
| DeiT | ViT-Ti/16 | RegNetY | 74.5 | 22.6 | 7.7 | 36.6 | 67.4 | 73.3 | 35.8 | 0.474 |
| Proteus | ViT-Ti/14 | DINOv2-S/14 | 75.2 | 26.7 | 11.3 | 39.8 | 67.3 | 80.2 | 40.7 | 0.423 |
| DeiT | ViT-S/16 | RegNetY | 81.2 | 32.0 | 20.7 | 46.5 | 54.6 | 77.8 | 42.5 | 0.445 |
| Proteus | ViT-S/14 | DINOv2-B/14 | 81.8 | 39.8 | 31.1 | 51.0 | 50.5 | 85.8 | 50.0 | 0.350 |
| DeiT | ViT-B/16 | RegNetY | 83.4 | 36.0 | 29.6 | 50.7 | 46.1 | 81.4 | 46.3 | 0.419 |
| Proteus | ViT-B/14 | DINOv2-L/14 | 84.9 | 51.0 | 52.4 | 63.4 | 38.6 | 88.6 | 54.4 | 0.303 |

**Comparison with Distillation in Supervised Learning.** As Proteus only requires ImageNet-1K for pre-training, we compare our method with the supervised training scheme DeiT (Touvron et al., 2021) which is equipped with knowledge distillation from multiple perspectives. In terms of ImageNet accuracy, the advantage of Proteus increases when we scale up the model and this phenomenon correlates with the trend in robustness evaluation on four ImageNet variants. We then validate the two approaches on 12 fine-grained classification datasets to test their generalization ability and Proteus consistently outperforms DeiT at different scales as we effectively transfer the fruitful knowledge from the foundation model DINOv2 (Oquab et al., 2023). Moreover, Proteus exhibits remarkable advantages in dense prediction tasks and the improvement also increases when we scale up the models. In conclusion, Proteus constantly outperforms the conventional supervised learning setting across various dimensions, offering a novel training scheme enhanced by foundation models.

## 3.3 COMPRESSING DINOv2

### 3.3.1 TARGET MODEL: VIT-S

In this part, we choose randomly initialized ViT-S (Dosovitskiy et al., 2020) with the patch size of 14 as the backbone following DINOv2 (Oquab et al., 2023) and utilize pre-trained DINOv2-B as the teacher network. Note that official DINOv2-S/B/L models are obtained by distilling from DINOv2-g which has stronger performance, it increases the training time significantly due to a very large teacher. Since the CLIP-series models (Radford et al., 2021; Cherti et al., 2023) do not offer ViT-S level options, we compare our method with ViT-B/32 level models, despite the latter having a larger number of parameters.

**ImageNet Linear Evaluation.** Following the design from DINOv2 (Oquab et al., 2023), we concatenate features from multiple layers for linear probing on ImageNet-1K and rerun all the baseline methods in this setup. Shown in Tab. 4, Proteus outperforms all the CLIP (Radford et al., 2021; Cherti et al., 2023) models on ImageNet-1K significantly and even surpasses the official DINOv2 model. The possible explanation is that we conduct pre-training on ImageNet-1K training set which shares a similar distribution with its validation set. However, it is noteworthy that Proteus also surpasses other self-supervised approaches pre-trained on ImageNet-1K (e.g., DINO (Caron et al., 2021) reaches 78.2% with ViT-B/16, and iBOT (Zhou et al., 2021) attains 81.0% with ViT-L/16). Additionally, Proteus outperforms supervised learning methods, such as DeiT (Touvron et al., 2021), which achieves 81.2% with ViT-S/16 when using knowledge distillation.

**Fine-grained Classification.** In Tab. 4, we validate the frozen features across 12 fine-grained classification benchmarks, in accordance with DINOv2 (Oquab et al., 2023). With only 1.2M images for pre-training, Proteus outperforms CLIP (WIT-400M) and OpenCLIP (LAION-400M) and achieves similar performance with OpenCLIP (LAION-2B). Despite the fact that a great number of classes in those 12 benchmarks do not correlate with ImageNet-1K, Proteus exhibits surprising generalization ability and it demonstrates that Proteus effectively transfers the fruitful knowledge from the teacher model DINOv2-B, which is trained on massive data. However, Proteus still lags behind DINOv2-S on FGVC-Aircraft, Stanford Cars, and SUN397 as its pre-training dataset, i.e., LVD-142M, is built by retrieving similar images with their training sets.

**Semantic Segmentation.** We consider two setups to evaluate the dense prediction ability of Proteus. *Linear*: we evaluate on the frozen patch features and only train a linear layer to predict the class logits (Zhou et al., 2021; Oquab et al., 2023). *Fine-tune*: we utilize UperNet (Xiao et al., 2018) as

Table 4: Comparison on ImageNet linear evaluation and fine-grained classification. Proteus, which is trained with only 1.2M images, outperforms CLIP (WIT-400M) and OpenCLIP (LAION-400M) and achieves comparable results with OpenCLIP (LAION-2B). †DINOv2-S is distilled from DINOv2-g while Proteus is distilled from DINOv2-B.

| Method | Arch | # imgs | ImageNet | Aircraft | Cal101 | Cars | C10 | C100 | DTD | Flowers | Food | Pets | SUN | VOC | CUB | Average |
|---|---|---|---|---|---|---|---|---|---|---|---|---|---|---|---|---|
| DINOv2† | ViT-S/14 | 142M | 81.1 | 74.4 | 96.5 | 80.8 | 97.7 | 87.7 | 80.3 | 99.5 | 89.2 | 95.1 | 74.5 | 88.0 | 87.5 | 87.6 |
| CLIP | ViT-B/32 | 400M | 74.7 | 50.1 | 93.6 | 81.5 | 95.1 | 80.2 | 76.7 | 96.6 | 88.5 | 89.3 | 76.6 | 87.2 | 75.2 | 82.5 |
| OpenCLIP | ViT-B/32 | 400M | 74.4 | 52.8 | 93.4 | 88.8 | 95.3 | 82.1 | 79.0 | 96.7 | 86.2 | 88.5 | 75.3 | 86.5 | 76.4 | 83.4 |
| OpenCLIP | ViT-B/32 | 2B | 76.7 | 59.1 | 95.3 | 92.2 | 96.8 | 86.0 | 81.7 | 97.6 | 87.9 | 90.6 | 77.7 | 87.3 | 77.7 | 85.8 |
| Proteus† | ViT-S/14 | 1.2M | 81.8 | 65.5 | 95.1 | 77.0 | 97.8 | 87.3 | 78.4 | 99.0 | 87.9 | 95.7 | 71.7 | 87.1 | 86.7 | 85.8 |

Table 5: Results on Semantic Segmentation dataset ADE20K using UperNet with 512×512 resolution. † use patch size of 14×14, thus adapt to the resolution of 518×518.

| Method | Arch | # imgs | mIoU (%) | |
|---|---|---|---|---|
| | | | Linear | Fine-tune |
| DINOv2† | ViT-S/14 | 142M | 45.7 | 51.0 |
| DeiT | ViT-S/16 | 1.2M | 29.1 | 42.5 |
| iBOT | ViT-S/16 | 1.2M | 30.4 | 45.5 |
| OpenCLIP | ViT-B/32 | 400M | 33.4 | 45.6 |
| OpenCLIP | ViT-B/32 | 2B | 35.7 | 47.6 |
| Proteus† | ViT-S/14 | 1.2M | 41.8 | 50.0 |

Table 6: Results on Depth Estimation dataset NYU-Depth V2 using DPT with 512×512 resolution. † use patch size of 14×14, thus adapt to the resolution of 518×518.

| Method | Arch | # imgs | RMSE↓ (%) | |
|---|---|---|---|---|
| | | | Linear | Fine-tune |
| DINOv2† | ViT-S/14 | 142M | 0.344 | 0.327 |
| DeiT | ViT-S/16 | 1.2M | 0.584 | 0.445 |
| iBOT | ViT-S/16 | 1.2M | 0.461 | 0.381 |
| OpenCLIP | ViT-B/32 | 400M | 0.450 | 0.390 |
| OpenCLIP | ViT-B/32 | 2B | 0.424 | 0.379 |
| Proteus† | ViT-S/14 | 1.2M | 0.404 | 0.350 |

the task layer and fine-tine the whole network (Tian et al., 2023; He et al., 2022). Tab. 5 shows that Proteus achieves similar performance with the Oracle method DINOv2 (Oquab et al., 2023) on full fine-tuning and outperforms other methods by a clear margin, despite that OpenCLIP (Cherti et al., 2023) is trained with tons of data and iBOT (Zhou et al., 2021) is trained with self-supervised patch-level learning objectives for 800 epochs. Under the linear evaluation protocol, DINOv2 achieves the strongest performance but Proteus still exhibits significant advantages over other baselines.

**Depth Estimation.** We validate our approach on the monocular depth estimation benchmark NYU-Depth V2 and adapt the same evaluation settings in Semantic Segmentation. Utilizing DPT (Ranftl et al., 2021) as the decoder, the phenomenon shown in Tab. 6 is similar with the one in Tab. 5, i.e., Proteus clearly outperforms other baseline methods but lags behind the Oracle method DINOv2 (Oquab et al., 2023) which is pre-trained on massive high-quality data with delicate learning objectives.

### 3.3.2 SCALING UP

In this part, we scale up our experiments by utilizing DINOv2-L/g (Oquab et al., 2023) as the teachers and compress them to ViT-B/L (Dosovitskiy et al., 2020) respectively to examine the scalability of Proteus. We compare our method with DINOv2 (Oquab et al., 2023), CLIP (Radford et al., 2021), OpenCLIP (Cherti et al., 2023), and SynCLR (Tian et al., 2023), which is trained on 600M synthetic data. The scaling behavior of Proteus is validated in both high-level understanding tasks (Tab. 7) and dense prediction tasks (Tab. 8 and Tab. 9).

**Classification.** For ImageNet linear evaluation, Proteus-B surpasses DINOv2-B and exhibits remarkable advantages over all the SynCLR and CLIP variants, including the ViT-Large models. When scaling up to ViT-L, DINOv2-L outperforms Proteus-L by 0.1% with a much larger training set. In terms of fine-grained classification results, Proteus-B outperforms ViT-Base level CLIP models and achieves similar average accuracy with their ViT-Large models. As SynCLR and DINOv2 manually include the distribution of these fine-grained datasets by generation or retrieval, Proteus-B obtains comparable performance with SynCLR-B and falls behind DINOv2-B by a small margin. It is worth noting that Proteus is only pre-trained on 1.2M images and we observe the discrepancy of average accuracy with the Oracle method DINOv2 decreases from 1.8% to 1.4% when we scale up from Small to Base. When further scaling up our method, Proteus-L demonstrates superior performance compared to SynCLR-L, and it even achieves results comparable to the Oracle method DINOv2-L, indicating the strong scaling capabilities of Proteus.

Table 7: Comparison on ImageNet linear evaluation and fine-grained classification with larger models. †DINOv2-B/L are distilled from DINOv2-g while Proteus-B/L are distilled from DINOv2-L/g.

| Method | Arch | # imgs | ImageNet | Aircraft | Cal101 | Cars | C10 | C100 | DTD | Flowers | Food | Pets | SUN | VOC | CUB | Average |
|---|---|---|---|---|---|---|---|---|---|---|---|---|---|---|---|---|
| DINOv2† | ViT-B/14 | 142M | 84.5 | 79.9 | 96.7 | 88.1 | 98.8 | 91.3 | 82.1 | 99.7 | 92.8 | 96.0 | 77.2 | 88.0 | 89.3 | 90.0 |
| | ViT-L/14 | 142M | 86.3 | 81.7 | 97.1 | 89.8 | 99.4 | 93.6 | 83.3 | 99.7 | 94.3 | 96.4 | 78.4 | 87.9 | 90.7 | 91.0 |
| CLIP | ViT-B/16 | 400M | 78.7 | 58.3 | 94.7 | 87.0 | 95.9 | 82.6 | 78.0 | 97.9 | 92.8 | 93.2 | 78.8 | 88.2 | 81.8 | 85.8 |
| | ViT-L/14 | 400M | 83.4 | 68.5 | 96.6 | 91.1 | 98.0 | 87.4 | 81.9 | 99.2 | 95.3 | 95.2 | 81.8 | 88.8 | 85.6 | 89.1 |
| OpenCLIP | ViT-B/16 | 400M | 78.7 | 59.1 | 95.7 | 92.1 | 96.4 | 84.0 | 81.7 | 98.2 | 90.7 | 91.7 | 78.2 | 87.9 | 82.9 | 86.5 |
| | ViT-L/14 | 400M | 81.7 | 65.2 | 96.7 | 93.9 | 97.9 | 88.0 | 82.8 | 98.8 | 93.4 | 93.3 | 80.9 | 88.5 | 86.9 | 88.8 |
| | ViT-L/14 | 2B | 81.9 | 69.5 | 96.3 | 95.0 | 97.7 | 87.9 | 84.1 | 98.9 | 92.6 | 93.8 | 81.9 | 88.9 | 85.6 | 89.3 |
| SynCLR | ViT-B/16 | 600M | 80.5 | 81.8 | 94.4 | 93.9 | 96.6 | 84.8 | 79.9 | 99.2 | 91.6 | 93.6 | 76.2 | 89.4 | 83.8 | 88.8 |
| | ViT-L/14 | 600M | 82.8 | 85.5 | 95.7 | 94.2 | 98.1 | 88.2 | 82.1 | 99.2 | 93.4 | 94.5 | 78.2 | 90.4 | 86.8 | 90.5 |
| Proteus† | ViT-B/14 | 1.2M | 84.9 | 72.9 | 96.1 | 83.7 | 98.8 | 90.6 | 81.4 | 99.4 | 91.6 | 96.2 | 75.5 | 88.4 | 88.3 | 88.6 |
| | ViT-L/14 | 1.2M | 86.2 | 83.6 | 96.8 | 89.1 | 99.3 | 93.5 | 83.4 | 99.6 | 93.9 | 96.8 | 77.3 | 88.9 | 90.2 | 91.0 |

Table 8: Results on Semantic Segmentation dataset ADE20K using UperNet with larger models at 518×518 resolution.

| Method | Arch | Teacher | # imgs | mIoU |
|---|---|---|---|---|
| DINOv2 | ViT-B/14 | DINOv2-g/14 | 142M | 55.8 |
| | ViT-L/14 | DINOv2-g/14 | 142M | 58.1 |
| Proteus | ViT-B/14 | DINOv2-L/14 | 1.2M | 54.4 |
| | ViT-L/14 | DINOv2-g/14 | 1.2M | 57.7 |

Table 9: Results on Depth Estimation dataset NYU-Depth V2 using DPT with larger models at 518×518 resolution.

| Method | Arch | Teacher | # imgs | RMSE↓ |
|---|---|---|---|---|
| DINOv2 | ViT-B/14 | DINOv2-g/14 | 142M | 0.281 |
| | ViT-L/14 | DINOv2-g/14 | 142M | 0.243 |
| Proteus | ViT-B/14 | DINOv2-L/14 | 1.2M | 0.304 |
| | ViT-L/14 | DINOv2-g/14 | 1.2M | 0.240 |

**Dense Prediction.** We further evaluate the scalability of Proteus at dense prediction tasks, including semantic segmentation and depth estimation. Here we only focus on the *Fine-tune* setup to strive for better performance. One can observe from Tab. 8 and Tab. 9 that the discrepancy between Proteus and the Oracle model DINOv2 decreases with the expanding model size. Even though Proteus is only trained with 0.8% of DINOv2's training data with different data distribution, Proteus-L even outperforms DINOv2-L in depth estimation task on the NYU-Depth V2 dataset, suggesting the effectiveness of our design.

## 3.4 GENERALIZATION TO OTHER APPROACHES

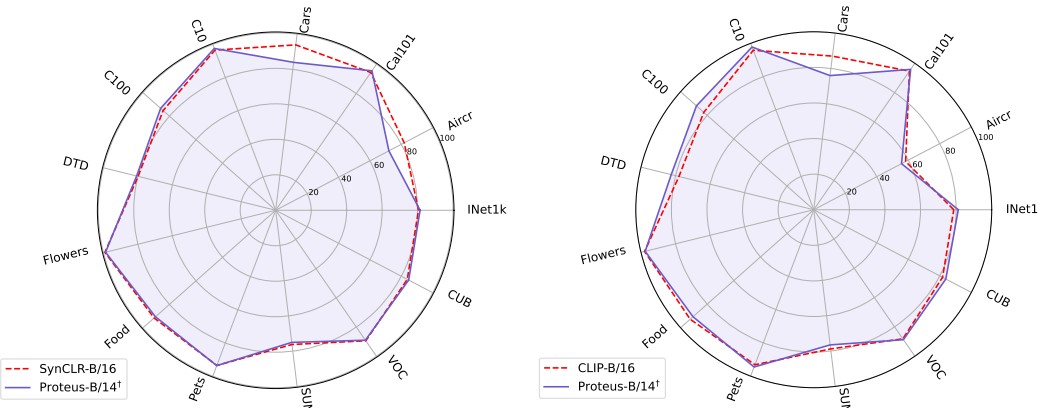

Figure 4: Comparison on ImageNet linear evaluation and fine-grained classification with SynCLR. †Proteus is distilled from SynCLR-L/14.

Figure 5: Comparison on ImageNet linear evaluation and fine-grained classification with CLIP. †Proteus is distilled from CLIP-L/14.

We test the generalization ability of Proteus by leveraging other foundation models SynCLR (Tian et al., 2023) and CLIP (Radford et al., 2021) as the teacher networks. SynCLR is trained with the contrastive learning objective on the undisclosed 600M synthetic dataset, while CLIP is obtained by aligning images and corresponding text descriptions through contrastive learning on the private dataset WIT-400M. We utilize SynCLR-L/14 and CLIP-L/14 as the teachers, and we remove the patch and feature learning objectives for CLIP training following the original design. Shown in

Table 10: Comparison on ImageNet linear evaluation and fine-grained classification with different proxy datasets. We follow the default protocol of Proteus to train ViT-S/14 distilled by DINOv2-B/14.

| Dataset | ImageNet | Aircraft | Cal101 | Cars | C10 | C100 | DTD | Flowers | Food | Pets | SUN | VOC | CUB | Average |
|---|---|---|---|---|---|---|---|---|---|---|---|---|---|---|
| ImageNet-1K | 81.8 | 65.5 | 95.1 | 77.0 | 97.8 | 87.3 | 78.4 | 99.0 | 87.9 | 95.7 | 71.7 | 87.1 | 86.7 | 85.8 |
| ImageNet-Merge | 81.8 | 68.5 | 95.3 | 79.9 | 98.1 | 87.1 | 79.6 | 99.0 | 91.1 | 95.5 | 72.9 | 87.5 | 86.4 | 86.7 |
| ImageNet-Single | 63.7 | 51.2 | 86.4 | 46.9 | 88.4 | 72.7 | 71.9 | 92.3 | 67.0 | 64.4 | 61.0 | 78.3 | 43.4 | 68.7 |

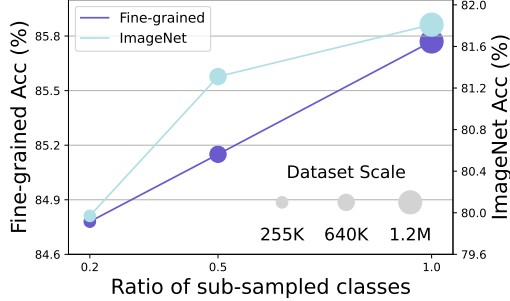

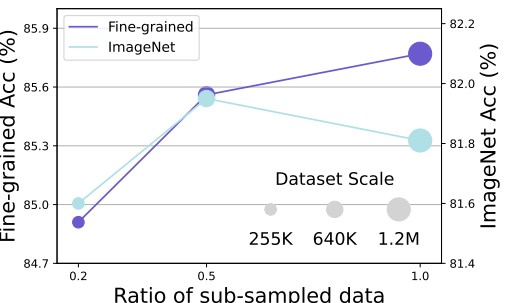

Figure 6: Comparisons on ImageNet linear evaluation and fine-grained classification on ImageNet-1K with sub-sampled classes.

Figure 7: Comparisons on ImageNet linear evaluation and fine-grained classification on ImageNet-1K with sub-sampled data per class.

Fig. 4, Proteus exceeds SynCLR at ImageNet linear evaluation (81.4% versus 80.5%) and falls behind on 12 fine-grained datasets (87.4% versus 88.8%). A similar phenomenon is observed in Fig. 5, where Proteus outperforms CLIP on ImageNet (81.2% versus 78.7%) and performs similarly at fine-grained classification (85.7% versus 85.8%). Beyond the accuracy comparisons, we notice that the distribution of Proteus's results is very similar to its counterparts, which are trained with different learning objectives on large-scale datasets under the scaling law. This validates the generalization ability of Proteus, i.e., *it is what it accesses*, as it effectively transfers the knowledge from various foundation models in a task-agnostic manner.

## 3.5 ANALYSIS ON PROXY DATASET

While ImageNet-1K serves as a good proxy dataset, we further study the importance of the proxy dataset by introducing multiple variants: (1) ImageNet-Merge: Following the idea of DINOv2 (Oquab et al., 2023), we simply concatenate all the training sets that we utilized for validation, including ImageNet-1K, 12 fine-grained datasets, ADE20K and NYU-Depth V2. It results in a training set of 1.4M images. (2) ImageNet-Single: We build a 1.2M training set where the images are generated from a single large image by performing extensive data augmentations (Asano & Saeed, 2021). (3) ImageNet-sub-data: we randomly sample a portion of data at each class. (4) ImageNet-sub-class: we randomly sample a portion of classes from the total 1000 classes. All ViT-S/14 models are distilled from DINOv2-B/14 for 375K iterations with a batch size of 1024. More details are in Sec. A.11.

**Dataset Diversity.** Shown in Tab. 10, ImageNet-Merge brings almost 1% average improvement on fine-grained classification with merely 0.2M additional data, which suggests the importance of a diverse dataset. We further consider the extreme case where all the source information comes from a single image and it surprisingly delivers very decent performance. Specifically, 6 out of 12 datasets suffer from performance drop over 10% while the remaining ones exhibit acceptable results considering that most of the classes in those fine-grained datasets are missing in this single image. This demonstrates that Proteus is robust even under the extreme scenario.

**Scaling Behavior.** Fig. 6 shows that there is a minor performance drop (1%) at fine-grained accuracy even when we only sample 200 classes out of 1000. More surprisingly, the ImageNet linear probing accuracy only decreases by 1.8% when we sample 200 classes (80% of the classes are not seen by the model during pre-training). Fig. 7 suggests a similar phenomenon but there are two differences: (1) sub-sampling data suffers from less performance drop which indicates that data diversity is more important than the number of data per class. (2) There is a 0.1% increase in ImageNet accuracy when sampling 50% of the data, which we hypothesize is due to the variance in linear probing. To

summarize, these two figures demonstrate the robustness of Proteus when subjected to reduced data availability, suggesting that it is feasible to access foundation models with even smaller data scales.

## 4    RELATED WORK

**Supervised Learning** approaches (He et al., 2016; Simonyan & Zisserman, 2014; Krizhevsky et al., 2012) used to dominate the research community of computer vision. These methods provide valuable experience in model architecture design, which benefit the research in different domains (Zhang et al., 2018a;b; Vaswani et al., 2017; Dosovitskiy et al., 2020). Nevertheless, the reliance on extensively annotated datasets (Deng et al., 2009; Lin et al., 2014; Geiger et al., 2012) inherent in supervised learning models presents significant limitations. Additionally, supervised learning can struggle with generalization outside of the training dataset (Koh et al., 2021; Recht et al., 2019), especially in cases where the distribution of the training data does not accurately reflect real-world scenarios.

**Self-supervised Learning** has emerged as a powerful paradigm for leveraging unlabeled data, significantly reducing the dependency on extensive labeled datasets. Contrastive learning methods (Chen et al., 2020; He et al., 2020; Oord et al., 2018; Tian et al., 2020; Hadsell et al., 2006) foster invariance to different transformations of the same image while segregating the representations of distinct images (Wang & Isola, 2020). On the other hand, masked image modeling (Bao et al., 2021; He et al., 2022; Wei et al., 2022; Zhou et al., 2021) involves either pixel reconstruction (He et al., 2022) or local feature regeneration (Baevski et al., 2022), often yielding impressive results when fine-tuning for dense prediction tasks. In the landscape of vision foundation models, strategies such as DINOv2 (Oquab et al., 2023) and CLIP (Radford et al., 2021) have set new benchmarks for versatility and robustness. However, both approaches conduct large-scale training on private gigantic datasets, presenting substantial challenges for training foundation models in a similar vein.

**Model Compression** techniques like Knowledge Distillation (Hinton et al., 2015; Li et al., 2020; Chen et al., 2019; Yin et al., 2020; Chen et al., 2022) and Pruning (Han et al., 2015; Li et al., 2016; Molchanov et al., 2016; Liu et al., 2018) have been extensively studied in recent years. While there are attempts to reduce the scale of training data, they mainly focus on the setting of task-specific model compression and lack examination on the generalization ability. However, task-agnostic model compression, which maintains the generalization ability, has become rather important in the era of foundation models (Radford et al., 2021; Oquab et al., 2023; Tian et al., 2023). Nevertheless, there remain two critical challenges: (1) The training data of those foundation models is rarely disclosed (Radford et al., 2021; Oquab et al., 2023; Tian et al., 2023). (2) Training those foundation models requires significantly large training resources and it is a common protocol to utilize the original dataset to perform task-agnostic model compression (Wu et al., 2023; Oquab et al., 2023). To address these issues, we target task-agnostic model compression with limited data.

## 5    DISCUSSION

**Limitation.** Due to computational constraints, we only validate our model on ImageNet-1K, and its efficacy at a larger scale dataset remains unverified. Besides, Proteus has to keep the same patch size as the teacher model because of the introduced patch and feature-level learning objectives.

**Broader Impact.** Discussed in Sec. 3.2, Proteus surpasses the supervised learning method across all metrics, demonstrating its potential as a new training paradigm. Besides, our work supports the research in model compression so that foundation models can be compressed to arbitrary scales at a much smaller cost. Shown in Sec. 3.5, it is possible to access foundation models at even smaller datasets compared to ImageNet-1K, which may be a promising avenue for future exploration. Finally, while our work mainly focuses on the pure vision foundation models, we hope our work can incentivize the exploration of this idea in other modalities (LMMs, VLMs) to facilitate the research.

**Conclusion.** In this paper, we propose *Proteus*, a simple and general distillation framework to transfer the general-purpose visual representations from foundation models into the target network with 'limited' data (ImageNet-1K). Specifically, we remove the designs from conventional knowledge distillation that could lead to dataset bias and present three levels of training objectives, i.e., token, patch, and features, to promise the application across various tasks. We conduct extensive experiments to verify the scalability and generalization ability of Proteus and provide comprehensive analysis.

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

# A APPENDIX

## A.1 IMPLEMENTATION DETAILS

### A.1.1 PRE-TRAINING

We strictly follow the DeiT (Touvron et al., 2021) training protocol on ImageNet-1K (Deng et al., 2009) except that we do not enable Mixup (Zhang et al., 2017). All models are training for 300 epochs with a batch size of 1024 on 8 A100 GPUs, except the ViT-L experiment with a batch size of 256 due to GPU memory constraints. We adopt the masking strategy in iBOT (Zhou et al., 2021) for the patch-level learning objective.

### A.1.2 IMAGENET LINEAR PROBING

For all ImageNet linear probing results, we follow the linear probing protocol outlined in DI-NOv2 (Oquab et al., 2023). Specifically, the linear layer is trained using the SGD optimizer for 25,020 iterations with a batch size of 512. We employ random-resized-crop data augmentation and conduct a grid search over the hyperparameters as defined in DINOv2. We report the best accuracy on the validation set in accordance with prior works.

### A.1.3 FINE-GRAINED LINEAR CLASSIFICATION

Following SynCLR (Tian et al., 2023), we train a regularized multinomial logistic regression model on the output classification token. During training and testing, no data augmentation is applied; images will be resized to 224 along the shorter side, followed by a center crop of 224×224. The training objective is minimized using L-BFGS with L2-regularization and the L2-regularization constant is chosen on the validation set from 45 logarithmically spaced values between $10^{-6}$ and $10^3$. Unlike DINOv2 (Oquab et al., 2023) and SynCLR, We reduce the maximum number of L-BFGS iterations from 1000 to 500 for faster evaluation.

### A.1.4 SEMANTIC SEGMENTATION

We conduct experiments on ADE20K dataset (Zhou et al., 2017) using the single-scale setup. We adopt the resolution of 512×512 for models that are trained with the patch size of 16×16 and 518×518 for patch size of 14×14. *Linear*: we follow the setup and hyperparameters in iBOT (Zhou et al., 2021) and use FCN (Long et al., 2015) as the linear layer to perform probing on the frozen backbone. *Fine-tune*: we adhere to the setup and hyperparameters in SynCLR (Tian et al., 2023) which utilizes UperNet (Xiao et al., 2018) as the task adaptation layer.

### A.1.5 DEPTH ESTIMATION

We evaluate the methods on the NYU-Depth V2 (Silberman et al., 2012) dataset. Similar with semantic segmentation, we utilize a resolution of 512×512 for models trained with a patch size of 16×16, and a resolution of 518×518 for models trained with a patch size of 14×14. We always adopt DPT (Ranftl et al., 2021) as the decoder under both evaluation protocols. *Fine-tune*: we follow to the setup and hyperparameters in the public codebase (Li, 2022). *Linear*: we utilize the same hyperparameters in the *Fine-tune* protocol except that we enlarge the learning rate from 1e-4 to 4e-4.

## A.2 WEIGHT INHERITANCE

Existing methods to compress foundation models often utilize the pre-trained weights either by structure pruning (Ma et al., 2023; Xia et al., 2023) or weight inheritance (Wu et al., 2023; Xu et al., 2023). However, structure pruning requires complex setups and is not general enough to be adapted to arbitrary architectures. Weight Selection (Xu et al., 2023) proves that initializing smaller models with uniformly selected weights from a pre-trained larger model results in better performance. Besides, TinyCLIP (Wu et al., 2023) shows simply inheriting the weights from the teacher network, i.e., the foundation model, with a fixed pattern can improve the distillation efficiency. Following the design of Weight Selection, we conduct ablation on DeiT (Touvron et al., 2021) and DINOv2 (Oquab et al., 2023), where the pre-trained weights are learned with supervised learning

Table 11: Ablation study of Weight Inheritance on DeiT where the model is trained with the standard protocol on ImageNet-1K for 300 epochs.

| Arch | Inherit | Teacher | ImageNet |
|---|---|---|---|
| DeiT-Ti | N/A | N/A | 71.9 |
| DeiT-Ti | DeiT-S | N/A | 73.5 |
| DeiT-Ti | N/A | DeiT-S | 73.7 |
| DeiT-Ti | DeiT-S | DeiT-S | 74.8 |

Table 12: Ablation study of Weight Inheritance on DINOv2 where the model is trained with the standard protocol on ImageNet-1K for 300 epochs.

| Arch | Inherit | Teacher | ImageNet |
|---|---|---|---|
| DINOv2-Ti | N/A | N/A | 75.4 |
| DINOv2-Ti | DINOv2-S | N/A | 75.8 |
| DINOv2-Ti | N/A | DINOv2-S | 77.5 |
| DINOv2-Ti | DINOv2-S | DINOv2-S | 77.2 |

objective and self-supervised learning objective, respectively. Table 11 demonstrates that weight inheritance is advantageous for DeiT training, whether knowledge distillation is applied or not. Conversely, as shown in Table 12, weight inheritance is less effective for DINOv2 and can even be detrimental when knowledge distillation is used. We conjecture that this phenomenon is caused by the mismatch between the downstream learning objective (supervised learning) and the pre-training learning objective (self-supervised learning), while both the teacher and student of DeiT are trained with the supervised learning objective.

To validate this point, we conduct ablation on weight inheritance and Tab. 13 shows that directly inheriting the pre-trained weights results in performance degradation at all classification benchmarks. Therefore, we simply leverage the randomly initialized network as the student to maintain the generalization ability of Proteus, so that it could be applied to different foundation models which are learned with various learning objectives.

Table 13: Comparison on ImageNet linear evaluation and fine-grained classification with ablation on weight inheritance. We follow the default protocol of Proteus to train ViT-S/14 with DINOv2-B/14.

| Method | Inherit | ImageNet | Aircraft | Cal101 | Cars | C10 | C100 | DTD | Flowers | Food | Pets | SUN | VOC | CUB | Average |
|---|---|---|---|---|---|---|---|---|---|---|---|---|---|---|---|
| Proteus | ✔ | 81.5 | 64.5 | 94.9 | 75.9 | 97.7 | 86.6 | 78.2 | 98.8 | 87.7 | 94.9 | 71.1 | 87.0 | 86.4 | 85.3 |
| Proteus | | 81.8 | 65.5 | 95.1 | 77.0 | 97.8 | 87.3 | 78.4 | 99.0 | 87.9 | 95.7 | 71.7 | 87.1 | 86.7 | 85.8 |

## A.3 ROBUSTNESS EVALUATION

We compare Proteus with DINOv2 on ImageNet variants for robustness evaluation. We utilize the models that are obtained by linear probing on ImageNet-1K and run inference on those datasets. Shown in Tab. 14, Proteus achieves similar results with DINOv2 on ImageNet-Sketch, ImageNet-R and outperforms it obviously on ImageNet-C. However, we fall behind DINOv2 at ImageNet-A at different scales which may be caused by the lack of training data.

Table 14: Comparison with DINOv2 on robustness evaluation.

| Method | Dataset | Arch | Teacher | Accuracy | Robustness | | | |
|---|---|---|---|---|---|---|---|---|
| | | | | Im-Val | Im-S | Im-A | Im-R | Im-C↓ |
| DINOv2 | LVD-142M | ViT-S/14 | DINOv2-g/14 | 81.2 | 40.2 | 34.1 | 53.4 | 54.3 |
| Proteus | ImageNet-1.2M | ViT-S/14 | DINOv2-B/14 | 81.8 | 39.8 | 31.1 | 51.0 | 50.5 |
| DINOv2 | LVD-142M | ViT-B/14 | DINOv2-g/14 | 83.4 | 50.5 | 55.5 | 63.9 | 42.8 |
| Proteus | ImageNet-1.2M | ViT-B/14 | DINOv2-L/14 | 84.9 | 51.0 | 52.4 | 63.4 | 38.6 |
| DINOv2 | LVD-142M | ViT-L/14 | DINOv2-g/14 | 86.3 | 59.4 | 71.4 | 74.6 | 31.4 |
| Proteus | ImageNet-1.2M | ViT-L/14 | DINOv2-g/14 | 86.2 | 59.8 | 68.7 | 73.9 | 29.9 |

## A.4 MORE DIVERSE PROXY DATASET FOR CLIP

While we have shown that Proteus achieves similar average accuracy on fine-grained classification with CLIP (Radford et al., 2021) when leveraging CLIP-L/14 as the teacher, it is worth noting that the performance at certain datasets, e.g., StanfordCars and FGVC-Aircraft, is obviously worse than

CLIP. We conjecture the limitation is caused by the diversity of ImageNet-1K so we reconduct the experiments with ImageNet-Merge. Shown in Tab.15, the accuracy of these datasets increases significantly, even outperforming CLIP on FGVC-Aircraft. This indicates that the performance of Proteus can be easily enhanced by a more diverse dataset.

Table 15: Comparison on fine-grained classification with more diverse proxy dataset. We distill CLIP-L/14 to ViT-B/14.

| Method | Dataset | Aircraft | Cal101 | Cars | C10 | C100 | DTD | Flowers | Food | Pets | SUN | VOC | CUB | Average |
|---|---|---|---|---|---|---|---|---|---|---|---|---|---|---|
| CLIP-B/16 | WIT-400M | 58.3 | 94.7 | 87.0 | 95.9 | 82.6 | 78.0 | 97.9 | 92.8 | 93.2 | 78.8 | 88.2 | 81.8 | 85.8 |
| Proteus-B/14 | ImageNet-1K | 55.7 | 95.9 | 76.0 | 97.8 | 88.1 | 82.5 | 97.6 | 90.7 | 94.5 | 76.5 | 88.7 | 83.8 | 85.7 |
| Proteus-B/14 | ImageNet-Merge | 62.8 | 96.4 | 86.8 | 97.9 | 88.6 | 81.8 | 98.4 | 94.1 | 94.2 | 78.8 | 88.8 | 84.9 | 87.8 |

## A.5 DENSE PREDICTION FOR SYNCLR

In the main text, we have compared Proteus with SynCLR (Tian et al., 2023) on the classification tasks when utilizing it as the teacher network. As SynCLR is also trained with the patch-level objective iBOT (Zhou et al., 2021), it exhibits strong performance on dense prediction tasks. We further fine-tune the pre-trained backbones on semantic segmentation dataset ADE20K (Zhou et al., 2017) and utilize UperNet (Xiao et al., 2018) as the task adaptation layer. Shown in Tab. 16, Proteus lags behind SynCLR only by 0.9% which is acceptable considering the small scale dataset that we use. It further demonstrates the generalization ability of Proteus as SynCLR is trained with a very different learning objective on the large synthetic dataset.

Table 16: Comparison on semantic segmentation dataset ADE20K with SynCLR. [†]Proteus is distilled from SynCLR ViT-L/14.

| Method | Arch | text | img | # imgs | mIoU |
|---|---|---|---|---|---|
| SynCLR | ViT-B/16 | synthetic | synthetic | 600M | 53.4 |
| Proteus[†] | ViT-B/14 | N/A | real | 1.2M | 52.5 |

## A.6 GENERALIZATION TO OUT-OF-DOMAIN CONCEPTS

Following SynCLR (Tian et al., 2023), we evaluate the pre-trained models on two additionally fine-grained datasets: GTSRB (Stallkamp et al., 2011) and Country211 (Radford et al., 2021). These two datasets are not in the retrieval list or concept list in DINOv2 and SynCLR, therefore considered as out-of-domain concepts. Tab. 17 shows that DINOv2 and SynCLR obtain obviously worse performance compared to CLIP, possibly because its training data included similar concepts. While Proteus achieves very similar results with its counterparts on both datasets by mimicking the teacher's pattern, indicating its strong generalization ability across different foundation models.

Table 17: Generalization to concepts not seen by DINOv2 and SynCLR.

| Method | Arch | Teacher | GTSRB | Country211 |
|---|---|---|---|---|
| DINOv2 | ViT-B/14 | DINOv2-g/14 | 73.5 | 21.8 |
| Proteus | ViT-B/14 | DINOv2-L/14 | 76.4 | 19.4 |
| SynCLR | ViT-B/16 | N/A | 78.5 | 21.0 |
| Proteus | ViT-B/14 | SynCLR-L/14 | 81.5 | 21.4 |
| CLIP | ViT-B/16 | N/A | 88.2 | 32.3 |
| Proteus | ViT-B/14 | CLIP-L/14 | 87.6 | 29.3 |

## A.7 TEACHER CHOICE

In the default setting of Proteus, we choose the pre-trained model at a larger scale to serve as our teacher network and we conduct ablation with teacher at different scales for analysis. Shown in Tab. 18, the average accuracy of Proteus decreases with the increased teacher scales. This may

seem counterintuitive as a larger teacher contains more fruitful knowledge and has stronger ability. However, there exists a dimension gap between teacher and student, and a projection head is utilized to align the dimensions between them. Assumably, a larger teacher denotes a lager gap between the dimension and information may be lost during the projection. This facilitates the application of Proteus because smaller teachers will speedup the training process significantly compared with giant teacher models.

Table 18: Comparison on fine-grained classification with different teacher choices. We follow the default protocol of Proteus to train ViT-S/14 with different teachers. $^{\dagger}$ denotes the official DINOv2-S model which is distilled from DINOv2-g.

| Teacher | Aircraft | Cal101 | Cars | C10 | C100 | DTD | Flowers | Food | Pets | SUN | VOC | CUB | Average |
|---|---|---|---|---|---|---|---|---|---|---|---|---|---|
| DINOv2-g/14$^{\dagger}$ | 74.4 | 96.5 | 80.8 | 97.7 | 87.7 | 80.3 | 99.5 | 89.2 | 95.1 | 74.5 | 88.0 | 87.5 | **87.6** |
| DINOv2-g/14 | 60.3 | 89.4 | 69.8 | 96.7 | 83.0 | 70.9 | 94.1 | 82.7 | 94.9 | 63.0 | 85.2 | 77.3 | 80.6 |
| DINOv2-L/14 | 58.0 | 93.3 | 67.7 | 97.4 | 84.7 | 74.1 | 96.3 | 84.0 | 95.4 | 67.6 | 86.9 | 80.9 | 82.2 |
| DINOv2-B/14 | 65.5 | 95.1 | 77.0 | 97.8 | 87.3 | 78.4 | 99.0 | 87.9 | 95.7 | 71.7 | 87.1 | 86.7 | 85.8 |

## A.8 Training Efficiency

We compare with the supervised learning method DeiT (Touvron et al., 2021) in terms of training efficiency as both methods are trained on ImageNet-1K for 300 epochs. We measure the time on the same platform, i.e., 8 A100 GPUs. Shown in Tab. 19, Proteus consumes similar training hours with DeiT as our training objectives are very easy. Moreover, we find the validation after each training epoch actually costs lots of time in DeiT. In contrast, we perform pre-training on ImageNet-1K and only conduct linear probing after the training, which can be finished within one hour.

Table 19: Ablation study on projector design. ViT-S/14 models are distilled from DINOv2-B on ImageNet-1K for 300 epochs.

| Method | Arch | Teacher | Training Hour | Validation Hour |
|---|---|---|---|---|
| DeiT | ViT-S/16 | RegNetY | 30 | 9 |
| Proteus | ViT-S/14 | DINOv2-B/14 | 30 | 1 |

## A.9 Projection Head Design

Here we conduct ablation on the design of the projection head which is used to align the dimension between teacher and student model. Tab. 20 shows that different combination results in similar performance but Layer Normalization (LN) (Ba et al., 2016) leads to slightly better accuracy in terms of the two metrics. We adopt this design in all our experiments without fine-tuning for better strategy.

Table 20: Ablation study on projector design. ViT-S/14 models are distilled from DINOv2-B on ImageNet-1K for 300 epochs.

| Design | ImageNet | Fine-grained |
|---|---|---|
| Linear+GELU | 81.8 | 84.9 |
| GELU+Linear | 81.5 | 85.3 |
| LN+Linear | 81.7 | **85.3** |

## A.10 Visualization Implementations

Following the procedure in DINOv2 (Oquab et al., 2023) and SynCLR (Tian et al., 2023), we perform PCA on the image patches from the same set and colorize them using their first three components. We resize the images to $518\times518$ to fit the patch size of 14 and the pictures are captured from the Internet.

## A.11 COMPOSITION OF PROXY DATASET

Shown in Tab. 21 and Fig. 8, we provide detailed information about the proxy datasets that we used in Sec. 3.5.

Table 21: Proxy Dataset composition.

| Dataset | # imgs | Source |
|---------|--------|--------|
| ImageNet-Merge | 1.4M | ImageNet-1K, ADE20K, NYUd, 12 fine-grained datasets |
| ImageNet-Single | 1.2M | Single Image (see Fig. 8) |
| ImageNet-sub-data (50%) | 640K | ImageNet-1K |
| ImageNet-sub-class (50%) | 640K | ImageNet-1K |
| ImageNet-sub-data (20%) | 255K | ImageNet-1K |
| ImageNet-sub-class (20%) | 255K | ImageNet-1K |

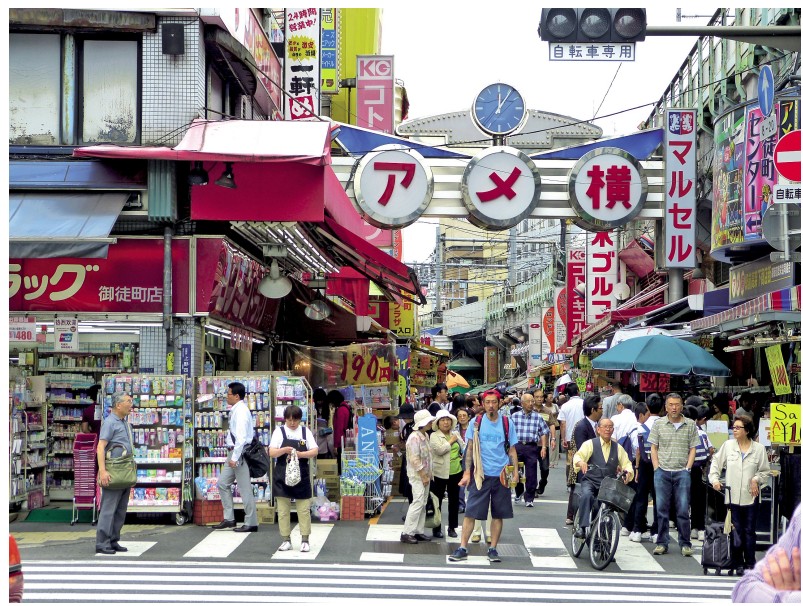

Figure 8: Picture that is used to generate ImageNet-Single.

## A.12 LIST OF BENCHMARKS FOR EVALUATION

We show the list of benchmarks used for evaluation in Tab. 22.

Table 22: List of benchmarks used for evaluation.

| Dataset name | Task | Citation |
|---|---|---|
| ImageNet-1K | Image Classification | (Deng et al., 2009) |
| ImageNet-A | Image Classification | (Hendrycks et al., 2021b) |
| ImageNet-C | Image Classification | (Hendrycks & Dietterich, 2019) |
| ImageNet-Rendition | Image Classification | (Hendrycks et al., 2021a) |
| ImageNet-Sketch | Image Classification | (Wang et al., 2019) |
| Food-101 | Image Classification | (Bossard et al., 2014) |
| CIFAR-10 | Image Classification | (Krizhevsky et al., 2009) |
| CIFAR-100 | Image Classification | (Krizhevsky et al., 2009) |
| SUN397 | Image Classification | (Xiao et al., 2010) |
| StanfordCars | Image Classification | (Krause et al., 2013) |
| FGVC-Aircraft | Image Classification | (Maji et al., 2013) |
| Pascal VOC 2007 | Image Classification | (Everingham et al., 2015) |
| Describable Textures | Image Classification | (Cimpoi et al., 2014) |
| Oxford Pets | Image Classification | (Parkhi et al., 2012) |
| Caltech101 | Image Classification | (Fei-Fei et al., 2004) |
| Oxford Flowers | Image Classification | (Nilsback & Zisserman, 2008) |
| CUB200 | Image Classification | (Wah et al., 2011) |
| GTSRB | Image Classification | (Stallkamp et al., 2011) |
| Country211 | Image Classification | (Radford et al., 2021) |
| ADE20K | Image Segmentation | (Zhou et al., 2017) |
| NYU-Depth V2 | Monocular Depth Estimation | (Silberman et al., 2012) |

