# OpenReview forum: "Accessing Vision Foundation Models via ImageNet-1K"
_ICLR.cc/2025/Conference — ICLR 2025 Poster_

### Official Review · Reviewer_Ktgp · 2024-10-19

**Soundness:** 3
**Presentation:** 3
**Contribution:** 3
**Rating:** 6
**Confidence:** 3

**Summary:**

The paper proposes Proteus, a method for distilling large vision foundation models into smaller equivalents on ImageNet-1K without access to the original training data. By focusing on token, patch, and feature-level training objectives, Proteus achieves impressive results at ImageNet-level costs, enabling broader access to training foundation models. When compared to other models including DINOv2-L/14, CLIP-L/14, OpenCLIP-L/14, and SynCLR-L/14, Proteus demonstrates competitive performance across 15 benchmarks, matching the Oracle method in some cases.

**Strengths:**

1. Proteus offers a practical and accessible method for distilling foundation models, addressing the resource-intensive nature of training such models.
2. Proteus demonstrates appealing performance and efficacy in knowledge transfer, matching or outperforming models trained on significantly larger datasets.
3. The paper is well-structured and effectively communicates the proposed solution and its results.

**Weaknesses:**

1. What is the performance of Proteus-L/14 distilled from the DINOv2-L/14 teacher? This informs the extent of performance degradation when distillation using ImageNet-1K.
2. Validation on varying data scales: The paper lacks validation on different data scales, such as subsets of ImageNet-1K or larger datasets like ImageNet-21K, which could provide more comprehensive insights.
3. Validation on other models and domains: The authors are suggested  to validate the effectiveness of  Proteus on diverse models and domains, such as large-scale pre-trained LLMs or diffusion models, to validate generalizability.

**Questions:**

Please refer to weaknesses.

---

> ### Author Response · Authors · 2024-11-24
> **reply to reviewer**
>
> We appreciate the Reviewer's approval and valuable comments. We respond to the Reviewer's concerns as below.
>
> ***
>
> **Weakness 1: Same-size teacher-student pair.**
>
> Thanks for the great suggestion. We conduct the experiment by utilizing DINOv2-S/14 as the teacher and Proteus-S/14 as the student to examine the effect and we choose smaller size networks considering the short time window. We validate the generalization ability of the distilled network by conducting linear probing on 12 fine-grained classification datasets.
>
> | Teacher  | Aircraft | Cal101 | Cars | C10 | C100 | DTD | Flowers | Food | Pets | SUN | VOC | CUB | Average |
> | :-----: | --- | --- | --- | --- | --- | --- | --- | --- | --- | --- | --- | --- | --- |
> | Oracle | 74.4% | 96.5% | 80.8% | 97.7% | 87.7% | 80.3% | 99.5% | 89.2% | 95.1% | 74.5% | 88.0% | 87.5% | 87.6% |
> | DINOv2-B | 65.5% | 95.1% | 77.0% | 97.8% | 87.3% | 78.4% | 99.0% | 87.9% | 95.7% | 71.7% | 87.1% | 86.7% | 85.8% |
> | DINOv2-S | 73.2% | 96.0% | 80.3% | 97.7% | 87.5% | 79.9% | 99.5% | 88.6% | 94.8% | 73.9% | 87.7% | 87.7% | 87.2% |
>
> We surprisingly find that we achieve almost the same performance with the oracle method DINOv2-S on the ViT-S architecture where our method exhibits the largest gap with DINOv2-S previously. This suggests that part of the performance gap is caused by the model capacity gap and our knowledge transfer is very efficient and effective. We will add this part in our final version.
>
> **Weakness 2: Varying data scales.**
>
> Thanks for the comment. We have provided the analysis of proxy dataset in Sec 3.5 of our paper and we examine the effect from two perspectives.
>
> (1) Dataset Diversity: we introduce ImageNet-Merge, which concatenates all the training sets that we utilized for validation, and ImageNet-Single, a 1.2M training set where the images are generated from a single large image by performing extensive data augmentations. We find ImageNet-Merge brings almost 1% average improvement on fine-grained classification with merely 0.2M additional data and ImageNet-Single surprisingly delivers very decent performance (68.7%), considering that all the information comes from a single picture. This demonstrates that Proteus is robust even under the extreme scenario.
>
> (2) Scaling Behavior: we conduct validation on subsets of ImageNet-1K by either sub-sampling a portion of data at each class or sub-sampling a portion of classes from the total 1000 classes. The results demonstrate the robustness of Proteus when subjected to reduced data availability, suggesting that it is feasible to access foundation models with even smaller data scales.
>
>
> **Weakness 3: Validation on other domains.**
>
> We thank the Reviewer for the valuable suggestion. Indeed, extending Proteus to LLMs and diffusion models is part of our original plan but we are struggling with computation resources as both of the experiments demand much larger training costs. We will definitely consider it as our future plan and we hope our work could incentivize the exploration of this idea in other modalities to facilitate the research.
>
> ***
>
> Considering the upcoming deadline of the discussion phase, we would appreciate it if the reviewer could provide any feedback based on our reply so that we could further improve our work. Thank you.

---

> > ### Comment · Reviewer_Ktgp · 2024-11-27
> >
> > Thank you for your rebuttal. I am happy to keep my supportive score.

---

> > > ### Author Response · Authors · 2024-12-01
> > > **thank you**
> > >
> > > We appreciate the reviewer's support and the valuable insights that helped us to improve the work. Thank you for being with us so far!

---

### Official Review · Reviewer_oEgL · 2024-11-01

**Soundness:** 3
**Presentation:** 3
**Contribution:** 3
**Rating:** 6
**Confidence:** 4

**Summary:**

This paper proposes a simple yet effective method for distilling the generalization ability of visual foundation models using only the ImageNet dataset. Specifically, compared to traditional knowledge distillation, it omits hard logits distillation, cross-entropy loss, and KL-divergence loss, while introducing MSE loss to align the intermediate results of the student network with those of the teacher network. The alignment of intermediate results involves three levels: token level, feature level, and patch level. Experiments show that the proposed method effectively distills the generalization ability of the teacher network.

**Strengths:**

1. This paper presents a simple yet effective method that utilizes the hidden layer features of the teacher network to successfully transfer its generalization ability to the student network.

2. The article provides extensive experimental validation of its claims on the ViT architecture, with datasets that include classification, segmentation, and depth estimation. The student network architecture encompasses various sizes, including ViT-small, ViT-base and ViT-large.

3. This paper is well-written.

**Weaknesses:**

1. This paper is less novelty. Previous works have also utilized intermediate layers for distillation on ImageNet-1K and achieved good performance on other datasets, such as MiniViT[1], and [2]. What are the differences between this work and those approaches. What are the comparative results of these methods with your work?

2. Figure 3 is not referenced in the main text, and it is unclear what issue Figure 3 is intended to illustrate.

3. When validating the generalization ability of the student network, how many epochs were fine-tuned for fine-grained classification, segmentation, and depth estimation? Compared to baseline methods like DeiT and IBOT, which used the same amount of data, was the computational load higher?

4. Why does Figure 7 show that more data leads to lower results on the ImageNet linear evaluation? For instance, sampling 50% of the data seems to yield better results than sampling 100% of the data.

5. How do the results of distillation when integrating hard logits distillation, the Cross-Entropy loss, the KL-divergence loss, and the MSE loss compare to the results obtained using only the proposed method? Would there be a decline in performance?

[1] Zhang J, Peng H, Wu K, et al. Minivit: Compressing vision transformers with weight multiplexing[C]//Proceedings of the IEEE/CVF Conference on Computer Vision and Pattern Recognition. 2022: 12145-12154.

[2] Zhang H, Zhou Y, Wang G H. Dense Vision Transformer Compression with Few Samples[C]//Proceedings of the IEEE/CVF Conference on Computer Vision and Pattern Recognition. 2024: 15825-15834.

**Questions:**

See weaknesses.

---

> ### Author Response · Authors · 2024-11-24
> **reply to reviewer**
>
> We sincerely appreciate the Reviewer’s detailed comments and constructive suggestions for us to improve our work. We make the response as below.
>
> ***
>
> **Weakness 1: Difference with prior arts.**
>
> We thank the Reviewer for pointing out these two methods, while we emphasize that we focus on the problem of data-free model compression in a task-agnostic context which is intrinsically different from them as we explained in the reply to all reviewers. Moreover, the mentioned approaches are more like pruning methods as they made hand-crafted designs on the architecture level while we did not touch the model structure.
>
> As there is no public implementation or pre-trained models of DC-ViT, we leverage the pre-trained model of MiniViT-B and examine its generalization ability by conducting linear probing on 12 fine-grained classification datasets. The results suggest that it performs on par with conventional knowledge distillation baseline DeiT-B and significantly lags behind our method Proteus-S. We will add the discussion with these methods in our final version.
>
> | Method  | Aircraft | Cal101 | Cars | C10 | C100 | DTD | Flowers | Food | Pets | SUN | VOC | CUB | Average |
> | :-----: | --- | --- | --- | --- | --- | --- | --- | --- | --- | --- | --- | --- | --- |
> | Proteus-S | 65.5% | 95.1% | 77.0% | 97.8% | 87.3% | 78.4% | 99.0% | 87.9% | 95.7% | 71.7% | 87.1% | 86.7% | 85.8% |
> | DeiT-B | 57.9% | 93.9% | 67.4% | 96.4% | 84.3% | 75.5% | 94.4% | 80.0% | 94.6% | 67.1% | 85.4% | 80.5% | 81.4% |
> | MiniViT-B | 61.6% | 92.4% | 66.1% | 96.0% | 82.5% | 74.4% | 92.3% | 79.6% | 93.1% | 64.8% | 85.8% | 76.7% | 80.4% |
>
>
> **Weakness 2: Figure 3.**
>
> Thanks for the great suggestion. We follow the same visualization protocol in DINOv2 where we compute a PCA between the patches of the images from the same set and colorize by their first 3 components. Figure 3 demonstrates the generalization ability and dense prediction ability of our method because we can accurately detect the foreground / background boundary as DINOv2, even though our training set is much smaller. We will add the description in the final version.
>
> **Weakness 3: Implementation details for downstream evaluation.**
>
> Thanks for the comment. The criterion of downstream evaluation is that we leverage the same setting for all methods, including training epochs, hyperparameters, etc. For fine-grained classification, we follow the setup in SynCLR where the training objective is minimized using L-BFGS with L2-regularization and we set the maximum iterations to 500. All methods are trained for 60K and 25K iterations for semantic segmentation and depth estimation, respectively.
>
> In terms of computational costs, we adopt the same architecture with DeiT-S/16 and iBOT-S/16 with the only difference in patch sizes as we have to keep the same patch size with our teacher network DINOv2-B/14 and our computational costs would be larger than DeiT-S/16 and iBOT-S/16. However, it is worth noting that Proteus-S/14 even outperforms DeiT-B/16 on fine-grained classification and dense prediction which suggests that our superior performance is not caused by the smaller patch size.
>
>
> **Weakness 4: Figure 7.**
>
> Thanks for reading our paper carefully. The reasons are two folds: (1) our method is very robust and sub-sampling data per class does not affect the performance significantly; (2) as we explained in the paper, we hypothesize the increase is due to the variance in linear probing as those models share very similar performance.
>
>
> **Weakness 5: Integrating logits distillation and cross-entropy loss.**
>
>
> We thank the Reviewer for the valuable suggestion. We conduct experiments by further integrating soft-logits distillation (it performs better than hard-logits distillation according to the empirical results) and cross-entropy loss to our framework (denoted as Proteus-S*) and we observe obvious performance decline in the generalization ability as these two designs introduce dataset bias again. We will add the results in our final version.
>
> | Method  | Aircraft | Cal101 | Cars | C10 | C100 | DTD | Flowers | Food | Pets | SUN | VOC | CUB | Average |
> | :-----: | --- | --- | --- | --- | --- | --- | --- | --- | --- | --- | --- | --- | --- |
> | Proteus-S | 65.5% | 95.1% | 77.0% | 97.8% | 87.3% | 78.4% | 99.0% | 87.9% | 95.7% | 71.7% | 87.1% | 86.7% | 85.8% |
> | Proteus-S* | 55.5% | 93.1% | 64.5% | 96.5% | 83.3% | 74.0% | 96.6% | 81.1% | 94.2% | 65.1% | 84.6% | 80.3% | 80.7% |
>
>
> ***
>
> Considering the upcoming deadline of the discussion phase, we would appreciate it if the reviewer could provide any feedback based on our reply so that we could further improve our work. Thank you.

---

> ### Comment · Reviewer_oEgL · 2024-11-27
>
> Thank you for your detailed and thoughtful response.
> 1. The authors clarify that distilling task-agnostic contextual information from the teacher model through data-free compression is a significant distinction from other knowledge distillation methods. If this is indeed the case, would similar phenomena be observed on datasets of comparable scale to ImageNet-1K? Furthermore, would the teacher model's generalization ability be more significantly affected on smaller or larger datasets? The experiments in the paper, which rely solely on variants of ImageNet-1K, do not sufficiently demonstrate the proposed method's dataset independence. The authors compare the downstream task performance of other transformer-based distillation methods, providing evidence of the effectiveness of the proposed approach in extracting generalization capabilities.
> 2. I understand the authors’ intended message conveyed by Figure 3. Including more similar results in the supplementary material would make the findings more convincing.
> 3. Comparisons on downstream tasks are fair when using identical training hyperparameters and iterations.
> 4. The authors' analysis does not seem to adequately explain the performance degradation observed when increasing the data scale by nearly half, as this counterintuitive result suggests an anomaly. The authors hypothesize that this is due to *variance in linear probing*. To substantiate this claim, results with finer-grained sampling rates—e.g., ratios of sub-sampled data at 0.5, 0.6, 0.7, 0.8, 0.9, and 1.0—would be more persuasive.
> 5. The authors provide additional experiments indicating that knowledge distillation using only intermediate results improves the student network's performance on downstream classification tasks. Including experiments on segmentation and depth estimation would further enhance the persuasiveness of the paper.
>
> Overall, leveraging smaller computational costs to extract the generalization capabilities of large-scale models for downstream tasks is a highly meaningful research direction. A smaller distillation dataset may cause the student model to quickly overfit to the distilled data, failing to extract generalization capabilities. Conversely, larger datasets may make learning challenging for the student model, requiring substantial computational cost to extract generalization. Balancing the scale of the distillation dataset and computational cost could be an intriguing research topic. The authors’ additional experiments validate the effectiveness of this approach in extracting transferability, and more experiments is necessary.

---

> > ### Author Response · Authors · 2024-12-01
> > **reply**
> >
> > We sincerely appreciate the reviewer’s valuable feedback, as well as the insightful suggestions that have helped us improve our work. We noticed that the reviewer has remaining questions and we would like to resolve the concerns by adding several key experiments before the end of the discussion phase:
> >
> > 1. Experiments on other datasets. While we have successfully replicated the performance of vision foundation models using ImageNet-1K as the training set, exploring alternative datasets should be viewed as a complementary direction as the results are influenced by various factors, including the number of samples, image quality, resolution, data augmentations, etc. Conducting a comprehensive study of this problem demands significant effort and resources. Considering the short time window, we introduce an additional dataset Conceptual Captions Dataset (CC3M) and download the images following the protocol[1].
> > To minimize resolution disparity with ImageNet data, we preserved the original aspect ratio of the images and set the shortest side to 384 pixels.
> > In the end, we managed to collect 2.2M images as different errors occurred during downloading (expired links, network issues, etc.) and we followed the default data augmentation operations used for ImageNet.
> > We conduct two experiments using DINOv2-B as the teacher and Proteus-S as the student: (1) randomly sampling 1.2M data from 2.2M images and keeping the original training epoch of 300; (2) leveraging 2.2M images as the training set and reducing the training epoch to 165 to match the same training time budget. The results in the table suggest that CC1.2M yields a marginal performance decline compared to ImageNet-1K, potentially due to differences in data quality and diversity. Interestingly, CC2.2M exhibits slightly inferior performance compared to CC1.2M, suggesting that scaling up data volume within a fixed training budget does not always translate to better results. Overall, our methods show competitive performance using web-sourced data which demonstrates the robustness of our approach and we hope these findings provide valuable insights for the research community.
> >
> > | Dataset  | Aircraft | Cal101 | Cars | C10 | C100 | DTD | Flowers | Food | Pets | SUN | VOC | CUB | Average |
> > | :-----: | --- | --- | --- | --- | --- | --- | --- | --- | --- | --- | --- | --- | --- |
> > | ImageNet-1K | 65.5% | 95.1% | 77.0% | 97.8% | 87.3% | 78.4% | 99.0% | 87.9% | 95.7% | 71.7% | 87.1% | 86.7% | 85.8% |
> > | CC1.2M | 64.2% | 94.4% | 73.8% | 96.7% | 83.8% | 79.2% | 99.2% | 86.5% | 94.0% | 74.0% | 87.1% | 80.3% | 84.4% |
> > | CC2.2M | 62.2% | 94.2% | 72.9% | 96.3% | 83.5% | 79.5% | 99.2% | 85.6% | 93.7% | 73.7% | 87.1% | 79.7% | 84.0% |
> >
> > 2. Yes, we will add more visualizations in the supplementary material in our final version.
> >
> > 4. Here are additional points we would like to highlight: (1) we maintain the same number of training iterations and batch sizes across all subset experiments, which means the training epochs for ImageNet-sub-data-50% are double those of the full dataset experiment, ensuring each model is trained for the same duration; (2) the accuracy gap between ImageNet-sub-data-50% (81.95%) and full dataset experiment(81.81%) is only 0.14%, while the gap between the ImageNet-sub-data-20% (81.60%) and full dataset experiment(81.81%) is only 0.21%. It suggests that our method is robust with various sampling ratios, though occasional "anomaly" may occur due to the inherent randomness in dataset creation and training processes.
> > We sincerely appreciate the reviewer’s insightful suggestion but providing results at such a fine-grained scale is currently infeasible due to the limited timeframe as the process involves iteratively creating datasets, conducting pre-training, and performing linear probing for different sampling ratios.
> > We will conduct this fine-grained analysis in our final version.
> >
> > 5. We further conduct experiments on semantic segmentation and depth estimation with both linear and fine-tune settings and the results suggest that introducing dataset bias results in obvious performance decline on downstream tasks as well.
> >
> > | Method  | mIOU (Linear) | mIOU (Fine-tune) | RMSE $\downarrow$ (Linear) | RMSE $\downarrow$ (Fine-tune) |
> > | :-----: | --- | --- | --- | --- |
> > | Proteus-S | 41.8% | 50.0% | 0.404% | 0.350% |
> > | Proteus-S* | 37.0% | 47.3% | 0.480% | 0.416% |
> >
> > We sincerely thank the reviewer for the detailed suggestions to improve our work and we will add these discussions in our final version. Please let us know if you have any remaining questions or concerns.
> >
> >
> > [1] https://github.com/rom1504/img2dataset/blob/main/dataset_examples/cc3m.md

---

> > > ### Author Response · Authors · 2024-12-02
> > > **reminder of deadline**
> > >
> > > Dear Reviewer oEgL,
> > >
> > > Thanks for your efforts so far in our paper review.
> > >
> > > Currently, most of the concerns from other Reviewers have been well-addressed and we are eager to know whether our response has resolved your concerns. Due to the coming discussion deadline (Dec 2nd), we would like to kindly remind Reviewer oEgL if our response through the whole rebuttal has helped the Reviewer to reevaluate this work.
> > >
> > > We really appreciate it if you could give us any feedback and your opinions are rather important to us. Thank you very much for your time!
> > >
> > > Sincerely,
> > > Authors

---

> > > ### Comment · Reviewer_oEgL · 2024-12-02
> > >
> > > Thank you for your reply. I have no further questions.

---

> > > > ### Author Response · Authors · 2024-12-02
> > > > **thank you**
> > > >
> > > > We appreciate the reviewer's support and the valuable insights that helped us to improve the work. Thank you for being with us so far!

---

### Official Review · Reviewer_tBLS · 2024-11-01

**Soundness:** 4
**Presentation:** 3
**Contribution:** 3
**Rating:** 8
**Confidence:** 5

**Summary:**

This paper aims to compress pre-trained vision foundation models, such as CLIP and DINO-V2, using only publicly available, small-scale datasets like ImageNet-1k, as the original training data is inaccessible.
To achieve this, this paper proposed _Proteus_, which perform knowledge distillation (KD) from the pre-trained model at token-level, patch-level, and feature-level. With extensive experiments, this paper demonstrate that the distilled model achieves comparable performance with the foundation model across multiple tasks.

**Strengths:**

1. This paper is well-motivated. With the development of foundation model trained with private data, it's difficult for the community to reproduce or compress the foundation model due to the inaccessible training data. Thus, how to use existing academic public datasets as a proxy to achieve comparable performance is an interesting question.

2. The performance of the model is excellent, as it achieve comparable performance with the foundation model with much less training data. Besides, this paper conducts extensive experiments and ablation studies to demonstrate the generalization ability of Proteus.

3. The empirically finding that conventional knowledge distillation with CE loss introduce dataset bias is interesting.

**Weaknesses:**

1. The novelty of this paper is limited. Although the results are promising, the loss at token, feature, and patch level has been widely explored in vision transformer distillation [1,2,3,4]. Besides, all of them can be considered as KD with intermediate features, which has been a common setting in KD. However, this paper does not cite and discuss any recent related works that perform KD with intermediate feature maps.

2. Although this paper presents experimental results to support that empirically removing CE loss can mitigate dataset bias, it would be better to provide some theoretical analysis to understand this question better.

[1] Yang, Zhendong, et al. "ViTKD: Feature-based Knowledge Distillation for Vision Transformers." Proceedings of the IEEE/CVF Conference on Computer Vision and Pattern Recognition. 2024.
[2] Chen, Xianing, et al. "Dearkd: data-efficient early knowledge distillation for vision transformers." Proceedings of the IEEE/CVF conference on computer vision and pattern recognition. 2022.
[3] Wang, Wenhui, et al. "Minilm: Deep self-attention distillation for task-agnostic compression of pre-trained transformers." Advances in Neural Information Processing Systems 33 (2020): 5776-5788.
[4] Wang, Wenhui, et al. "Minilmv2: Multi-head self-attention relation distillation for compressing pretrained transformers." arXiv preprint arXiv:2012.15828 (2020).

**Questions:**

The definition of "feature" is ambiguous. Currently, this paper just defines it as "intermediate feature _v_" (Line 171) without any additional information. This paper should specify which specific feature maps of vision transformers is used, like the output of the last transformer encoder layer of ViTs, the outputs of all 12 transformer encoder layers of ViTs, the attention maps, the Q/K/V within self-attention modules, etc.

---

> ### Author Response · Authors · 2024-11-24
> **reply to reviewer**
>
> We appreciate the Reviewer's feedback. We provide further explanations to clarify the Reviewer's concerns based on several key points as below.
>
> ***
>
> **Weakness 1: Comparison with prior arts.**
>
> We thank the Reviewer for pointing out these methods, while we emphasize that we focus on the problem of data-free model compression in a task-agnostic context which is intrinsically different from them as we explained in the reply to all reviewers.
>
> The first two methods (ViT-KD and DeadKD) are conventional knowledge distillation methods which leverage projection head and Cross-Entropy loss. These designs will result in dataset bias and hurt the generalization ability of student network based on our results in Table 1.
>
> MiniLM and MiniLMv2 also focus on the problem of task-agnostic model compression but they leverage the same training set as the teacher network, which aligns with the traditional setting in knowledge distillation and is fundamentally different from ours. To examine their effect in our setting, we implement MiniLMv2, which shows better performance compared to MiniLM, based on the public codebase and we validate the generalization ability of the student network on 12 fine-grained classification datasets. MiniLMv2 exhibits obviously inferior performance compared to our method, demonstrating that the problem we target to resolve is indeed challenging. We will add the discussion with these methods in our final version.
>
>
> | Method  | Aircraft | Cal101 | Cars | C10 | C100 | DTD | Flowers | Food | Pets | SUN | VOC | CUB | Average |
> | :-----: | --- | --- | --- | --- | --- | --- | --- | --- | --- | --- | --- | --- | --- |
> | Proteus-S | 65.5% | 95.1% | 77.0% | 97.8% | 87.3% | 78.4% | 99.0% | 87.9% | 95.7% | 71.7% | 87.1% | 86.7% | 85.8% |
> | MiniLMv2-S | 53.1% | 87.8% | 45.2% | 92.5% | 76.7% | 72.1% | 93.4% | 76.5% | 83.4% | 59.7% | 82.3% | 63.4% | 73.8% |
>
>
> **Weakness 2: Theoretical analysis.**
>
> Thanks for the great suggestion. While we agree that this is an important and intriguing phenomenon, providing a strict theoretical analysis is challenging due to the dynamic nature of the training process in knowledge distillation. However, we have attempted to provide an intuitive theoretical explanation based on the interaction between the CE loss and MSE loss: the CE loss forces the student model to fit the ImageNet-1K labels, aligning its feature space to the distribution of ImageNet-1K $P(T_{train})$, while the MSE loss aligns the student's features to the teacher's features, which are learned from a much larger and diverse dataset $P(T_{teacher})$.
>
> When CE loss is included, the student model's feature space is pulled toward the ImageNet-1K distribution, which introduces dataset bias and may limit its ability to generalize to other datasets $P(T_{test})$, as $P(T_{test})$ may share very different distributions with $P(T_{train})$, e.g., the 12 fine-grained classification datasets
> (conventional distillation methods mainly focus on the experiments on ImageNet-1K, so $P(T_{train})$ and $P(T_{test})$ are the training and validation sets of ImageNet-1K which are well aligned).
> While we have the prior knowledge that foundation models have great generalization ability, i.e., $P(T_{teacher})$ aligns better with $P(T_{test})$ than $P(T_{train})$, solely focusing on minimizing the discrepancy between the teacher and student features will allow the student model to better inherit the teacher’s generalization ability.
>
>
> **Question 1: Definition of 'feature'.**
>
> We thank the Reviewer for the valuable suggestion. The 'feature' represents the output of the last transformer layer in ViTs and we will add this detail in our final version.
>
>
> ***
>
> Considering the upcoming deadline of the discussion phase, we would appreciate it if the reviewer could provide any feedback based on our reply so that we could further improve our work. Thank you.

---

> ### Comment · Reviewer_tBLS · 2024-11-27
> **Thanks for your detailed response**
>
> I sincerely thanks the authors providing detailed responses, which solves my concerns in the weaknesses and questions .
>
> Moreover, the "Reply to all Reviewers" section clearly solves my major concern (i.e., the novelty) for this paper. From my personal perspective, the fundamental research question of this paper is actually **Knowledge Distillation with limited data**. This is an intriguing yet underexplored question, since more and more foundation models are developed with large-scale while unaccessible datasets. Previous works have focused solely on relatively simple tasks [1][2][3][4]. In contrast, this paper firstly explored this interesting research question on vision foundation model level and demonstrate the distilled model with limited data (comparing the scale of ImageNet to LAION, etc) can still maintain the original generalization ability. Considering that the setting itself is sufficiently novel, I believe the proposed token-, feature-, and patch-level distillation serves as a solid baseline in this direction. On the other hand, I would kindly suggest the authors avoid using "data-free", as it typically refers to scenarios involving learning with synthetic data [5][6].
>
> Therefore, I am revising my rating to positive in support of this paper. But this paper should do the following things in the final version:
> 1) emphasize more on the novel setting itself;
> 1) discuss more related works in terms of data-free KD, few-shot KD, etc;
> 2) clearly define the "features"
>
>
>
>
> [1] Li, Tianhong, et al. "Few sample knowledge distillation for efficient network compression." Proceedings of the IEEE/CVF conference on computer vision and pattern recognition. 2020.
>
> [2] Sauer, Anna, Shima Asaadi, and Fabian Küch. "Knowledge distillation meets few-shot learning: An approach for few-shot intent classification within and across domains." Proceedings of the 4th Workshop on NLP for Conversational AI. 2022.
>
> [3] Kang, Myeonginn, and Seokho Kang. "Knowledge distillation with insufficient training data for regression." Engineering Applications of Artificial Intelligence 132 (2024): 108001.
>
> [4] Cui, Kaiwen, et al. "KD-DLGAN: Data limited image generation via knowledge distillation." Proceedings of the IEEE/CVF Conference on Computer Vision and Pattern Recognition. 2023.
>
> [5] Chen, Hanting, et al. "Data-free learning of student networks." Proceedings of the IEEE/CVF international conference on computer vision. 2019.
>
> [6] Yin, Hongxu, et al. "Dreaming to distill: Data-free knowledge transfer via deepinversion." Proceedings of the IEEE/CVF conference on computer vision and pattern recognition. 2020.

---

> > ### Author Response · Authors · 2024-12-01
> > **thank you**
> >
> > We appreciate the reviewer's support and the valuable insights that helped us to improve the work. Yes, we will definitely revise those parts in our final version and add more discussions with those related works. Thank you for being with us so far!

---

### Official Review · Reviewer_cepE · 2024-11-01

**Soundness:** 3
**Presentation:** 3
**Contribution:** 3
**Rating:** 6
**Confidence:** 5

**Summary:**

While many Vision Foundation Model (VFM) checkpoints are available, their training datasets are inaccessible. This paper shows that distilling a vision foundation model using only the ImageNet dataset gives us a model that matches the performance of a similar size model trained with massive data, on several vision benchmark datasets from different domains. The paper uses the popular intermediate feature distillation strategy for KD. Experiments were conducted using several teacher-student combinations.

**Strengths:**

Presentation:
The paper was written well and was easy to follow.

Experiments:
Authors evaluated their models on a wide range of CV tasks (classification, segmentation, depth estimation) using several downstream datasets. Experimental results show that models distilled only on ImageNet using the proposed approach match/outperform same size models trained on significantly larger datasets without distillation. Experiments also show that the feature-based distillation performs better than logit-based distillation approach DeiT.

**Weaknesses:**

``The problem setting of the paper is not convincing``
While the original datasets used to train models such as DINOV2 are unavailable, there are numerous large scale image datasets (DataComp, LAION, etc.) that are available to the community these days. If the goal is to get a small model that performs well on a broad array of tasks, I do not understand why one should restrict themselves to distilling on ImageNet. They can use more diverse datasets, for example, diverse subsets of data taken from one of the large-scale public datasets. If the paper had compared different datasets in terms of their KD effectiveness or proposed methods to create better general purpose KD datasets, that would have been an interesting and useful contribution to the community.

The authors performed an experiment by using target task datasets for distillation and showed improved performance. This is an obvious/expected behavior, which also suggests that "we should go beyond ImageNet". If one has access to some data from target tasks, they can curate better distillation datasets as shown in a recent paper: "Knowledge Transfer from Vision Foundation Models for Efficient Training of Small Task-specific Models", ICML 2024.



``Limited technical contribution``
Feature distillation is a standard approach in ML community. There are numerous papers that proposed various distillation objectives based on intermediate features. See "Knowledge distillation: A survey", International Journal of Computer Vision, 2021.
The presented approach uses standard l2 distillation loss on classification and patch tokens. It leverages masking strategy which is also a commonly-used data augmentation approach when training vision models.


``Presentation - Redundant/Confusing terminology``
The terminology of token-level, feature-level and patch-level is a bit confusing/redundant. In a vision transformer setting each image patch becomes a token, so it is confusing to refer to "classification token-based loss" as "token-level" and "patch token-based loss" as "feature-level". Both losses are token-based losses (either CLS token or patch token). Alternatively, both losses can also be called feature-based losses (CLS token features or patch token features). Both feature-level and patch-level losses described here are based on patch-tokens. So, the terminology of "feature-level vs patch-level" is confusing since both are actually patch-level losses, one with masking and one without masking.

Table 1 - What does "Hint" in Table.1 refer to?

**Questions:**

Suggestions:

Training small models with distillation for improved performance is a fairly standard practice, and there are numerous distillation loss functions proposed over the last decade. An important question worth looking into in the current "data-centric ML world" is ``What is the right dataset to use for distillation?``. While the paper is looking into this, focusing solely on ImageNet dataset makes the paper's contribution fairly limited, especially given that there are several large-scale public image datasets available now.

You could avoid the redundant/confusing terminology used for the three loss terms in the paper.

---

> ### Author Response · Authors · 2024-11-24
> **reply to reviewer**
>
> We appreciate the Reviewer's feedback. We provide further explanations to clarify the Reviewer's concerns based on several key points as below.
>
> ***
>
> **Strengths 2: Experiments.**
>
> **'Experimental results show that models distilled only on ImageNet using the proposed approach match/outperform same size models trained on significantly larger datasets without distillation.'**
>
> We thank the Reviewer for noticing this phenomenon but we stress that the oracle models of DINOv2-based experiments (teacher networks) are obtained by distillation from DINOv2-g on their private large-scale datasets.
>
> **Weakness 1: Problem setting.**
>
> As pointed out in the paper mentioned by the reviewer ("Knowledge Transfer from Vision Foundation Models for Efficient Training of Small Task-specific Models"), pre-training on large-scale datasets requires significantly more computing which may not be affordable to most researchers, especially the ones in academia. ImageNet-1K has long been the cornerstone for advancements in computer vision and can be accessed by most researchers due to its relatively ‘small’ scale. Since our method has shown the ability to match the performance of oracle models in different tasks and datasets, we believe leveraging a smaller dataset to compress foundation models should be praised instead of criticized as our solution can be accessed by more audiences.
>
> **If the paper had compared different datasets in terms of their KD effectiveness or proposed methods to create better general purpose KD datasets, that would have been an interesting and useful contribution to the community.**
>
> We have provided the analysis of proxy dataset in Sec 3.5 of our paper and we examine the effect from two perspectives.
>
> (1) Dataset Diversity: we introduce ImageNet-Merge, which concatenates all the training sets that we utilized for validation, and ImageNet-Single, a 1.2M training set where the images are generated from a single large image by performing extensive data augmentations. We find ImageNet-Merge brings almost 1% average improvement on fine-grained classification with merely 0.2M additional data and ImageNet-Single surprisingly delivers very decent performance (68.7%), considering that all the information comes from a single picture. This demonstrates that Proteus is robust even under the extreme scenario.
>
> (2) Scaling Behavior: we conduct validation on subsets of ImageNet-1K by either sub-sampling a portion of data at each class or sub-sampling a portion of classes from the total 1000 classes. The results demonstrate the robustness of Proteus when subjected to reduced data availability, suggesting that it is feasible to access foundation models with even smaller data scales.
>
> **Weakness 2: Technical contribution.**
>
> While there are different kinds of feature distillation methods, we emphasize that we focus on the problem of data-free model compression in a task-agnostic context which is intrinsically different from them as we explained in the reply to all reviewers. Moreover, empirical results shown in that table demonstrate that directly adapting these methods to our problem does not work well and the problem we target to resolve is indeed challenging.
>
> **Weakness 3: Presentation.**
>
> We thank the Reviewer for the valuable suggestion and we will revise the terminology in our final version.
>
> **Table 1 - What does "Hint" in Table.1 refer to?**
>
> Hint refers to feature-based distillation according to the definition in FITNETS[1].
>
> ***
>
> Considering the upcoming deadline of the discussion phase, we would appreciate it if the reviewer could provide any feedback based on our reply so that we could further improve our work. Thank you.
>
> ***
>
> [1] Romero A, Ballas N, Kahou S E, et al. Fitnets: Hints for thin deep nets[J]. arXiv preprint arXiv:1412.6550, 2014.

---

> ### Comment · Reviewer_cepE · 2024-11-25
> **Thank you for the response.**
>
> I thank the authors for their detailed response.
>
> First, I would like to emphasize that while the training datasets used to train some proprietary foundation models are not available to public, there are various alternative large-scale datasets that are available for everyone (including academia) to use. For example, DataComp, LAION, etc. Hence, one need not restrict themselves to ImageNet.
>
> I understand that training on web-scale datasets may not be easy in academia. The reason I find the setting unconvincing is not about the data scale itself, but more about restricting only to ImageNet dataset. ImageNet has about 1.3M training images and the paper uses 300 epochs of training. That is equivalent to seeing around 400M examples during training. Since the goal is to get a small foundation model that works on a wide variety of tasks, it seems restrictive to focus only on ImageNet dataset. One can sample diverse subsets of data from existing web-scale datasets and use them for distillation within the same computational budget of "400M seen training samples". I think a thorough comparison of various alternative distillation datasets derived from existing large-scale computer vision datasets would make the paper more useful to the community.

---

> > ### Author Response · Authors · 2024-11-26
> > **reply**
> >
> > We sincerely appreciate the Reviewer’s prompt and valuable feedback, as well as the insightful suggestions that have helped us improve our work.
> >
> > However, we would like to clarify several points based on the remaining concerns:
> >
> > (1) 300-epoch training on ImageNet: we follow the training strategy of the supervised learning method DeiT[1] which is widely regarded as a de-facto standard training protocol in this area. **It suggests that we share similar training costs with those supervised learning methods, while our method offers significant performance boosts across various tasks, demonstrating its potential as a new training paradigm.**
> >
> > (2) Restricting to ImageNet: while we do agree with the reviewer that 'a thorough comparison of various alternative distillation datasets derived from existing large-scale computer vision datasets would make the paper more useful to the community', we believe such an exploration requires significant additional effort and is beyond the scope of this work. **The main contribution of our work is addressing the problem of task-agnostic model compression for vision foundation models—a challenge that has not been tackled before due to the reliance on private datasets and the substantial training costs—through the formulation of knowledge distillation to pursue a more general and accessible design.** While we have already demonstrated that our method matches the performance of foundation models using ImageNet, identifying a more representative large-scale proxy dataset should be considered a complementary direction rather than a core requirement. Moreover, we believe using smaller datasets better pronounces the effectiveness of our design as training on gigantic datasets like LAION-400M may make this problem fall into the conventional knowledge distillation setting, i.e., teacher and student networks share the same training set. Such a setup is less challenging and its efficacy remains unverified.
> >
> > (3) Not only ImageNet: while we did not explicitly investigate the effect of leveraging large-scale training sets, **we have analyzed the efficacy of proxy datasets in Section 3.5 where we introduce multiple choices of proxy dataset and the results suggest that it is feasible to access foundation models with even smaller data scales. This further validates the effectiveness of our method and provides valuable insights for the research community.**
> >
> > ***
> >
> > We appreciate the Reviewer's efforts so far for us to improve the work and we would be happy to address any further questions or concerns.
> >
> >
> > [1] Touvron H, Cord M, Douze M, et al. Training data-efficient image transformers & distillation through attention[C]//International conference on machine learning. PMLR, 2021: 10347-10357.

---

> > > ### Author Response · Authors · 2024-12-01
> > > **additional datasets**
> > >
> > > We noticed that the reviewers have remaining concerns about introducing additional datasets and we would like to resolve the concerns by adding several key experiments before the end of the discussion phase:
> > >
> > > While we have successfully replicated the performance of vision foundation models using ImageNet-1K as the training set, exploring alternative datasets should be viewed as a complementary direction as the results are influenced by various factors, including the number of samples, image quality, resolution, data augmentations, etc. Conducting a comprehensive study of this problem demands significant effort and resources. Considering the short time window, we introduce an additional dataset Conceptual Captions Dataset (CC3M) and download the images following the protocol[1].
> > > To minimize resolution disparity with ImageNet data, we preserved the original aspect ratio of the images and set the shortest side to 384 pixels.
> > > In the end, we managed to collect 2.2M images as different errors occurred during downloading (expired links, network issues, etc.) and we followed the default data augmentation operations used for ImageNet.
> > >
> > >
> > > We conduct two experiments using DINOv2-B as the teacher and Proteus-S as the student: (1) randomly sampling 1.2M data from 2.2M images and keeping the original training epoch of 300; (2) leveraging 2.2M images as the training set and reducing the training epoch to 165 to match the same training time budget. The results in the table suggest that CC1.2M yields a marginal performance decline compared to ImageNet-1K, potentially due to differences in data quality and diversity. Interestingly, CC2.2M exhibits slightly inferior performance compared to CC1.2M, suggesting that scaling up data volume within a fixed training budget does not always translate to better results. Overall, our methods show competitive performance using web-sourced data which demonstrates the robustness of our approach and we hope these findings provide valuable insights for the research community.
> > >
> > >
> > > | Dataset  | Aircraft | Cal101 | Cars | C10 | C100 | DTD | Flowers | Food | Pets | SUN | VOC | CUB | Average |
> > > | :-----: | --- | --- | --- | --- | --- | --- | --- | --- | --- | --- | --- | --- | --- |
> > > | ImageNet-1K | 65.5% | 95.1% | 77.0% | 97.8% | 87.3% | 78.4% | 99.0% | 87.9% | 95.7% | 71.7% | 87.1% | 86.7% | 85.8% |
> > > | CC1.2M | 64.2% | 94.4% | 73.8% | 96.7% | 83.8% | 79.2% | 99.2% | 86.5% | 94.0% | 74.0% | 87.1% | 80.3% | 84.4% |
> > > | CC2.2M | 62.2% | 94.2% | 72.9% | 96.3% | 83.5% | 79.5% | 99.2% | 85.6% | 93.7% | 73.7% | 87.1% | 79.7% | 84.0% |
> > >
> > >
> > > We sincerely thank the reviewer for the detailed suggestions to improve our work and we will add these discussions in our final version. Please let us know if you have any remaining questions or concerns.
> > >
> > >
> > > [1] https://github.com/rom1504/img2dataset/blob/main/dataset_examples/cc3m.md

---

> > > > ### Author Response · Authors · 2024-12-02
> > > > **reminder of deadline**
> > > >
> > > > Dear Reviewer cepE,
> > > >
> > > > Thanks for your efforts so far in our paper review.
> > > >
> > > > Currently, most of the concerns from other Reviewers have been well-addressed and we are eager to know whether our response has resolved your concerns. Due to the coming discussion deadline (Dec 2nd), we would like to kindly remind Reviewer cepE if our response through the whole rebuttal has helped the Reviewer to reevaluate this work.
> > > >
> > > > We really appreciate it if you could give us any feedback and your opinions are rather important to us. Thank you very much for your time!
> > > >
> > > > Sincerely,
> > > > Authors

---

> > > > ### Comment · Reviewer_cepE · 2024-12-02
> > > > **Thank you for additional experiments.**
> > > >
> > > > I thank the reviewer for providing additional experimental results. After reading the author’s response and other reviews, I have upgraded my rating.

---

> ### Author Response · Authors · 2024-12-02
> **thank you**
>
> We appreciate the reviewer's support and the valuable insights that helped us to improve the work. Thank you for being with us so far!

---

### Author Response · Authors · 2024-11-24
**reply to all reviewers**

Dear Reviewers:

Thanks for your valuable comments in the review process. As raised by most reviewers, we address the concern of 'limited novelty' from three perspectives


1. To the best of our knowledge, **we are the first to replicate the success of vision foundation models with ImageNet-1K as the training set** and experiments across various scales, tasks, datasets demonstrate the generalization ability of our delivered models. **This would benefit the research community as it eliminates the dependence on undisclosed datasets and significantly reduces the training costs associated with large-scale datasets, facilitating the accessibility of training foundation models for the broader research community.** In other words, we now have the ability to compress/train vision foundation models like DINOv2 to any scale that will satisfy the need, rather than being restricted to the limited configurations provided by the original authors.

2. Technically, we focus on the problem of data-free model compression in a task-agnostic context which is intrinsically different with conventional distillation methods from two perspectives. **(1) Data-free model compression**: our goal is to compress the foundation models, which are trained with tons of private data, on a much smaller dataset while maintaining its generalization ability. This problem is non-trivial as conventional knowledge distillation on ImageNet-1K (including the hint-based distillation methods) [1,2,3,4,5] leverage the projection head and Cross-Entropy loss which will result in dataset bias and hurt the generalization ability of student network based on our results in Table 1. **(2) Task-agnostic model compression**: most foundation models are trained with self-supervised learning objectives which promise their application on dense prediction tasks as well. This also distinguishes our method from prior arts as we aim to develop a uniform learning objective but they mainly consider the classification-level learning objective, which will limit their performance on dense prediction tasks which is shown in Table 2. **We have also add comparisons with a few distillation methods mentioned by the reviewers and the results in the table below suggests that directly adapting these methods to our problem does not work well.**

| Method  | Aircraft | Cal101 | Cars | C10 | C100 | DTD | Flowers | Food | Pets | SUN | VOC | CUB | Average |
| :-----: | --- | --- | --- | --- | --- | --- | --- | --- | --- | --- | --- | --- | --- |
| Proteus-S | 65.5% | 95.1% | 77.0% | 97.8% | 87.3% | 78.4% | 99.0% | 87.9% | 95.7% | 71.7% | 87.1% | 86.7% | 85.8% |
| MiniViT-B | 61.6% | 92.4% | 66.1% | 96.0% | 82.5% | 74.4% | 92.3% | 79.6% | 93.1% | 64.8% | 85.8% | 76.7% | 80.4% |
| MiniLMv2-S | 53.1% | 87.8% | 45.2% | 92.5% | 76.7% | 72.1% | 93.4% | 76.5% | 83.4% | 59.7% | 82.3% | 63.4% | 73.8% |

3. We admit that our method is simple, but that is the reason why **our method can compress different foundation models, trained with different self-supervised learning objectives and data sources, with a uniform design (shown in Section 3.4).** Moreover, **our analysis on the proxy dataset (shown in Section 3.5) suggests that it is possible to access foundation models at even smaller datasets compared to ImageNet-1K which further proves the robustness and effectiveness of our design.**


We thank the Reviewers for the constructive suggestions which help us to improve the work. Considering the upcoming deadline of the discussion phase, we would appreciate it if the reviewer could provide any feedback based on our reply so that we could further improve our work. Thank you so much for being with us so far.

Sincerely,
Authors


***

[1] Hinton G. Distilling the Knowledge in a Neural Network[J]. arXiv preprint arXiv:1503.02531, 2015.
[2] Romero A, Ballas N, Kahou S E, et al. Fitnets: Hints for thin deep nets[J]. arXiv preprint arXiv:1412.6550, 2014.
[3] Touvron H, Cord M, Douze M, et al. Training data-efficient image transformers & distillation through attention[C]//International conference on machine learning. PMLR, 2021: 10347-10357.
[4] Yang Z, Li Z, Zeng A, et al. ViTKD: Feature-based Knowledge Distillation for Vision Transformers[C]//Proceedings of the IEEE/CVF Conference on Computer Vision and Pattern Recognition. 2024: 1379-1388.
[5] Chen X, Cao Q, Zhong Y, et al. Dearkd: data-efficient early knowledge distillation for vision transformers[C]//Proceedings of the IEEE/CVF conference on computer vision and pattern recognition. 2022: 12052-12062.

---

> ### Comment · Reviewer_cepE · 2024-11-25
> **Disagree with the claim of "Data-free model compression"**
>
> I understand that the proposed strategy is task-agnostic since task-specific data is not used for distillation. But, I do not understand why it is data-free. Distillation is performed using ImageNet dataset, i.e., data is used for distillation.  If the authors were using some weight cloning/selection strategies to get the compressed model, the term "data-free" would have been meaningful. But, calling it data-free when performing standard data-based distillation is not appropriate.

---

> > ### Author Response · Authors · 2024-11-26
> > **reply**
> >
> > Thank you for pointing out the ambiguous phrasing. We use the term "data-free" [1,2] to emphasize that we do not have access to the original training set of the teacher network. Rather than relying on synthetic data or adversarial learning as substitutes for real data, we adopt ImageNet-1K as our proxy dataset to tackle this challenging problem.
> >
> >
> > ***
> >
> > [1] Lopes R G, Fenu S, Starner T. Data-free knowledge distillation for deep neural networks[J]. arXiv preprint arXiv:1710.07535, 2017.
> > [2] Chawla A, Yin H, Molchanov P, et al. Data-free knowledge distillation for object detection[C]//Proceedings of the IEEE/CVF Winter Conference on Applications of Computer Vision. 2021: 3289-3298.
> > [3] Luo L, Sandler M, Lin Z, et al. Large-scale generative data-free distillation[J]. arXiv preprint arXiv:2012.05578, 2020.

---

> ### Author Response · Authors · 2024-12-01
> **additional datasets**
>
> We noticed that some reviewers have remaining concerns about introducing additional datasets and we would like to resolve the concerns by adding several key experiments before the end of the discussion phase:
>
> While we have successfully replicated the performance of vision foundation models using ImageNet-1K as the training set, exploring alternative datasets should be viewed as a complementary direction as the results are influenced by various factors, including the number of samples, image quality, resolution, data augmentations, etc. Conducting a comprehensive study of this problem demands significant effort and resources. Considering the short time window, we introduce an additional dataset Conceptual Captions Dataset (CC3M) and download the images following the protocol[1].
> To minimize resolution disparity with ImageNet data, we preserved the original aspect ratio of the images and set the shortest side to 384 pixels.
> In the end, we managed to collect 2.2M images as different errors occurred during downloading (expired links, network issues, etc.) and we followed the default data augmentation operations used for ImageNet.
>
>
> We conduct two experiments using DINOv2-B as the teacher and Proteus-S as the student: (1) randomly sampling 1.2M data from 2.2M images and keeping the original training epoch of 300; (2) leveraging 2.2M images as the training set and reducing the training epoch to 165 to match the same training time budget. The results in the table suggest that CC1.2M yields a marginal performance decline compared to ImageNet-1K, potentially due to differences in data quality and diversity. Interestingly, CC2.2M exhibits slightly inferior performance compared to CC1.2M, suggesting that scaling up data volume within a fixed training budget does not always translate to better results. Overall, our methods show competitive performance using web-sourced data which demonstrates the robustness of our approach and we hope these findings provide valuable insights for the research community.
>
>
> | Dataset  | Aircraft | Cal101 | Cars | C10 | C100 | DTD | Flowers | Food | Pets | SUN | VOC | CUB | Average |
> | :-----: | --- | --- | --- | --- | --- | --- | --- | --- | --- | --- | --- | --- | --- |
> | ImageNet-1K | 65.5% | 95.1% | 77.0% | 97.8% | 87.3% | 78.4% | 99.0% | 87.9% | 95.7% | 71.7% | 87.1% | 86.7% | 85.8% |
> | CC1.2M | 64.2% | 94.4% | 73.8% | 96.7% | 83.8% | 79.2% | 99.2% | 86.5% | 94.0% | 74.0% | 87.1% | 80.3% | 84.4% |
> | CC2.2M | 62.2% | 94.2% | 72.9% | 96.3% | 83.5% | 79.5% | 99.2% | 85.6% | 93.7% | 73.7% | 87.1% | 79.7% | 84.0% |
>
>
> We sincerely thank the reviewer for the detailed suggestions to improve our work and we will add these discussions in our final version. Please let us know if you have any remaining questions or concerns.
>
> [1] https://github.com/rom1504/img2dataset/blob/main/dataset_examples/cc3m.md

---

### Meta-Review · Area_Chair_oJ1D · 2024-12-20

**Metareview:**

This paper proposed a new training recipe that enables distilling pretrained vision foundation models, e.g., CLIP and DINO-v2, into smaller-size models using only publicly available data such as ImageNet-1K. The recipe includes knowledge distillation at multiple levels, e.g., token, patch, and feature. Experiments show that the distilled model achieves comparable performance with the teacher model across multiple tasks.

The rebuttal has addressed the concerns of the reviewers about the novelty, technical contributions, and ablation experiments.

Although the technical contribution of knowledge distillation approach is not that significant, this work provides a valuable solution and baselines for training state-of-the-art vision foundation models at a relatively low cost. It allows and encourages more researchers in the community to explore and contribute.
Thus, I recommend to accept this paper.

**Additional Comments On Reviewer Discussion:**

The initial reviews are relatively negative with scores of 5, 5, 5, 6.
Reviewers raised concerns about limited novelty, unconvincing problem setting, limited technical contributions, and more ablation experiments.

In the rebuttal period, the authors clarified and emphasized the novelty in the literature and the significance of replicating vision foundation models at low costs via distillation. More ablation and comparison experiments were provided.

After rebuttal, three reviewers increased their ratings, resulting in to the final ratings of 6, 8, 6, 6.
All reviewers are positive about the paper.

---

### Decision · Program_Chairs · 2025-01-22

Accept (Poster)